# Pupil size modulation drives retinal activity in mice and shapes human perception

Tjasa Lapanja [1,2,11], Pietro Micheli[1,3,11], Andrés González-Guerra [1], Oleksandr Radomskyi[1], Gioia De Franceschi[1,4], Anna Muraveva [1], Alexander Attinger [5], Chiara Nina Roth[5], Matteo Tripodi[1], Tom Boissonnet[1], Marina Sabbadini[1,2,6], Josephine Jüttner [7], Petri Ala-Laurila[8,9], Georg Keller [5,10], Gabriel Peinado Allina [8], Hiroki Asari [1] & Santiago B. Rompani [1] ✉

Retinal adaptation is assisted by the pupil, with pupil contraction and dilation thought to prevent global light changes from triggering neuronal activity in the retina. However, we find that pupillary constriction from increased light, the pupillary light reflex (PLR), can drive strong responses in retinal ganglion cells (RGCs) in vivo in mice. The PLR drives neural activity in all RGC types, and pupil-driven activity is relayed to the visual cortex. Furthermore, the consensual PLR allows one eye to respond to luminance changes presented to the other eye, leading to a binocular response and modulation during low-amplitude luminance changes. To test if pupil-induced activity is consciously perceived, we performed psychophysics on human volunteers, finding a perceptual dimming consistent with PLR-induced responses in mice. Our findings thus uncover that pupillary dynamics can directly induce visual activity that is consciously detectable, suggesting an active role for the pupil in encoding perceived ambient luminance.

Visual perception relies on the detection of light and the subsequent processing of information by the retina and brain. However, modulation of visual input starts before the retina, at the pupil. The pupil can change its size based on the luminance of the environment, contrast, state of arousal, or mental effort, thereby affecting the amount of light reaching the retina[1–5]. The pupil is dilated in dim light, maximizing the number of photons entering the eye, while increased light causes the pupil to constrict via the pupillary light reflex (PLR)[6,7]. Pupillary constriction is thought to assist in normalizing photoreceptor activity to global changes in light levels[8,9], thus allowing the retina to keep

responding to spatially localized changes in luminance associated with visual features[10–12]. Furthermore, in bright ambient luminance, a small pupil is predicted to enhance contrast and reduce blur by directing the light through the central part of the lens, thus avoiding scatter caused by peripheral lens aberrations[13–15]. Counterintuitively, however, the pupil does not remain in a fully constricted state in daylight conditions. Rather, it maintains complex patterns of dilation and constriction[16–18], with the function of such pupillary dynamics remaining unclear.

Pupillary constriction via the PLR can reduce the amount of light reaching the retina by up to 10-fold, but the effect of this luminance

[1]Epigenetics and Neurobiology Unit, EMBL Rome, European Molecular Biology Laboratory, Monterotondo 00015, Italy. [2]Collaboration for Joint PhD Degree Between EMBL and Heidelberg University, Faculty of Biosciences, Heidelberg University, Heidelberg, Germany. [3]Department of Neuroscience, Collaboration for Joint PhD Degree Between EMBL Sapienza University, Rome, Italy. [4]KU Leuven, Department of Biology & Leuven Brain Institute, Leuven 3000, Belgium VIB, Leuven, Belgium. [5]Friedrich Miescher Institute for Biomedical Research, Basel, Switzerland. [6]Cell Biology and Biophysics Unit, European Molecular Biology Laboratory (EMBL), Heidelberg, Germany. [7]Institute of Molecular and Clinical Ophthalmology Basel, Basel, Switzerland. [8]Department of Neuroscience and Biomedical Engineering, Aalto University, Espoo, Finland. [9]Faculty of Biological and Environmental Sciences, Molecular and Integrative Biosciences Research Programme, University of Helsinki, Helsinki, Finland. [10]Faculty of Science, University of Basel, Basel, Switzerland. [11]These authors contributed equally: Tjasa Lapanja, Pietro Micheli. ✉e-mail: santiago.rompani@embl.it

change on retinal output is presumed to be negligible[13,19–21]. In this work, however, we explored whether pupillary dynamics could directly affect the neural activity of the retina by generating detectable transients in the overall amount of light entering the eye. Particularly, we hypothesized that pupillary constriction induced by the PLR during visual stimulation in daylight conditions could generate a sharp decrease in luminance on the retina, thus driving a global increase of activity in the OFF channel while decreasing the activity of the ON channel.

Pupillary constriction is induced by a subcortical loop that integrates luminance information from both eyes. As a result, pupillary constriction is consensual, with illumination of one eye leading to constriction of both pupils[21–24]. Additionally, binocular stimulation leads to increased pupil constriction compared to monocular stimulation[23]. Consequently, PLR-driven responses would lead to a non-canonical form of binocularity in the retina[24]. The retina is presumed to be monocular, with no binocular activity having ever been reported, to our knowledge, in the retinal ganglion cells (RGCs), the output cells of the retina. Binocular integration is widely described in areas of the visual pathway where monocular inputs can converge, such as the primary visual cortex (V1) or the dorsal lateral geniculate nucleus (dLGN). Specifically, neurons are defined as having binocular responses if stimulating either eye independently can elicit a neuronal response, and having binocular modulation if the other eye can either facilitate or suppress one eye's response[25,26]. Should the PLR lead to retinal activity, we predicted that the following binocular phenomena should also be seen in RGCs: (1) Stimulation of one eye would shape visual information on the other eye via the consensual PLR, consequently driving retinal activity. Such retinal neurons would, by definition, thus exhibit binocular responses. (2) Stimulation of both eyes would lead to a larger constriction than stimulating only one eye and, consequently, a larger luminance transient. Retinal neurons would thus exhibit binocular modulation. Finding these two types of responses would imply that the consensual pupillary response induces a non-canonical form of binocularity already at the retina.

Many visual processes that are output by the retina are subsequently corrected for before being allowed to impact conscious perception. For instance, saccades are able to elicit strong retinal responses, but those responses are not detected in the visual cortex or further downstream areas due to corollary discharge being able to inhibit such motion-induced responses[27,28]. A PLR-induced response could also be similarly inhibited, either in cortex or further downstream visual areas. Therefore, such activity should also be tested in the cortex and, ultimately, determined if it is available for conscious detection.

To test whether pupillary constriction can induce monocular and binocular retinal activity, we performed head-fixed in vivo calcium imaging of RGC axons in the dLGN. We indeed found that the PLR can induce monocular and binocular retinal responses, that pupil-mediated activity is strong, and consistently in both the OFF and ON channels. Pupil-induced binocular RGC responses can be facilitated by the interaction between direct and consensual PLR happening when the stimulus is presented to both eyes. Furthermore, we found that PLR-induced responses are detectable in the dLGN axons projecting to V1 and neurons in layer 2/3 of V1. Lastly, changes in luminance that in mice allow for PLR-induced RGC activity produces an illusory luminance change detectable by humans, suggesting such non-canonical retinal stimulation is available to conscious perception.

## Results

### Recording RGC boutons in the dLGN in vivo
To detect visually-evoked activity in the RGC axons in the dLGN in vivo, we injected an RGC-specific AAV[29] expressing GCaMP into a single eye (the right eye), then performed a craniotomy to access the dorsal surface of the left dLGN with 2-photon microscopy (Fig. 1A). This allowed for calcium imaging of dLGN-projecting RGCs axons (Fig. 1A-C) in anesthetized and awake mice[30,31], with all recordings in subsequent figures being anesthetized unless mentioned they are in awake. We presented visual stimulation on two curved screens with a custom-made separator between the eyes, allowing for independent stimulation of either the left (ipsilateral to imaged dLGN) or right (contralateral, GCaMP expressing) eye. The mice were first adapted to a gray background to both eyes. Next, we presented a series of full-field stimuli to either eye or both eyes in a pseudo-randomized order (see supplementary methods). Importantly, the non-stimulated eye received constant background stimulation.

### RGCs respond to direct and consensual pupillary constriction
We predicted that pupillary constriction during the full-field stimulus presentation could drive visual responses by producing a negative luminance transient on the retina. These pupillary-driven visual responses would occur after the onset of the stimulus, but with a delay consistent with the dynamic of pupillary constriction (Fig. 1D). Both ON and OFF RGCs could respond to pupillary-driven luminance transient. However, the response characteristics of OFF RGCs would allow for easier separation of canonical and pupil-driven responses. While the canonical response of OFF RGCs occurs at the offset of a full-field flash (Fig. 1D left, Fig. 1E left), a PLR-driven response would occur during the stimulation onset (Fig. 1D middle, Fig. 1F left). As the PLR acts consensually, stimulation of the ipsilateral eye would also lead to constriction of the contralateral pupil and, therefore, to a pupil-driven visual response in the absence of direct stimulation (Fig. 1D right, Fig. 1F right). This would suggest that the retina can produce a non-canonical binocular response by leveraging the PLR. Importantly, since these binocular responses would ultimately be the result of the luminance transient on the non-stimulated eye induced by the consensual PLR, they would only occur if the non-stimulated eye is not in complete darkness, but rather viewing a constant background.

OFF RGC boutons in the dLGN responded to full-field flashes presented to the contralateral (GCaMP expressing) eye with a canonical response to the offset of the visual stimulation (Figs. 2A–F and S1). Additionally, a substantial number of OFF boutons exhibited a delayed response consistent with pupillary dynamics in both anesthetized (55.6% of OFF boutons ($n = 361/649$ N = 5 animals)) and awake animals (30.5% of OFF boutons ($n = 216/709$ N = 4 animals)) (Fig. 2A-H, purple traces, S1D). ON boutons, on the other hand, showed a negative modulation of the response to contralateral stimulation during PLR (Fig. S1A-B, $n = 693$ boutons, N = 5 animals). As predicted by the hypothesis of consensual PLR driving binocular responses, stimulation of the ipsilateral (non-recorded) eye led to a similarly delayed visual response during the flash in 45.9% of OFF RGCs ($n = 298/649$ N = 5 animals) (Fig. 2A–F, green traces). This binocular visual response driven by ipsilateral stimulation was substantial, reaching on average 64.2% of the maximal response to contralateral stimulation (0.79 dF/F and 1.23 dF/F, respectively). We refer to the non-canonical response component of OFF RGCs during the flash as the "non-canonical OFF response". ON boutons also exhibited, albeit weaker, responses to ipsilateral stimulation (Figs. S1A–C and S2). As we hypothesized, these binocular responses were absent when the contralateral eye was in complete darkness, emphasizing the role of background luminance in enabling non-canonical responses (Fig. S3, $n = 572$ boutons, N = 3 animals).

Pupillary constriction started $0.51 \pm 0.16$ s after the onset of the stimulation, reaching peak constriction $1.74 \pm 0.37$ s after stimulus onset (Figs. 2G and S4E-F), correlating with the non-canonical RGCs responses (Figs. 2I and S1F). While the consensual PLR was generally weaker than direct PLR, the overall dynamics of constriction, both in terms of time to onset and time to peak was not significantly different (Figs. 2G and S1E, F), as was previously reported[21]. We did, however, observe animal to animal variability in the strength of PLR (Fig. S4A–C, $n = 849$ boutons, N = 5 animals), likely due to the effect of anesthesia

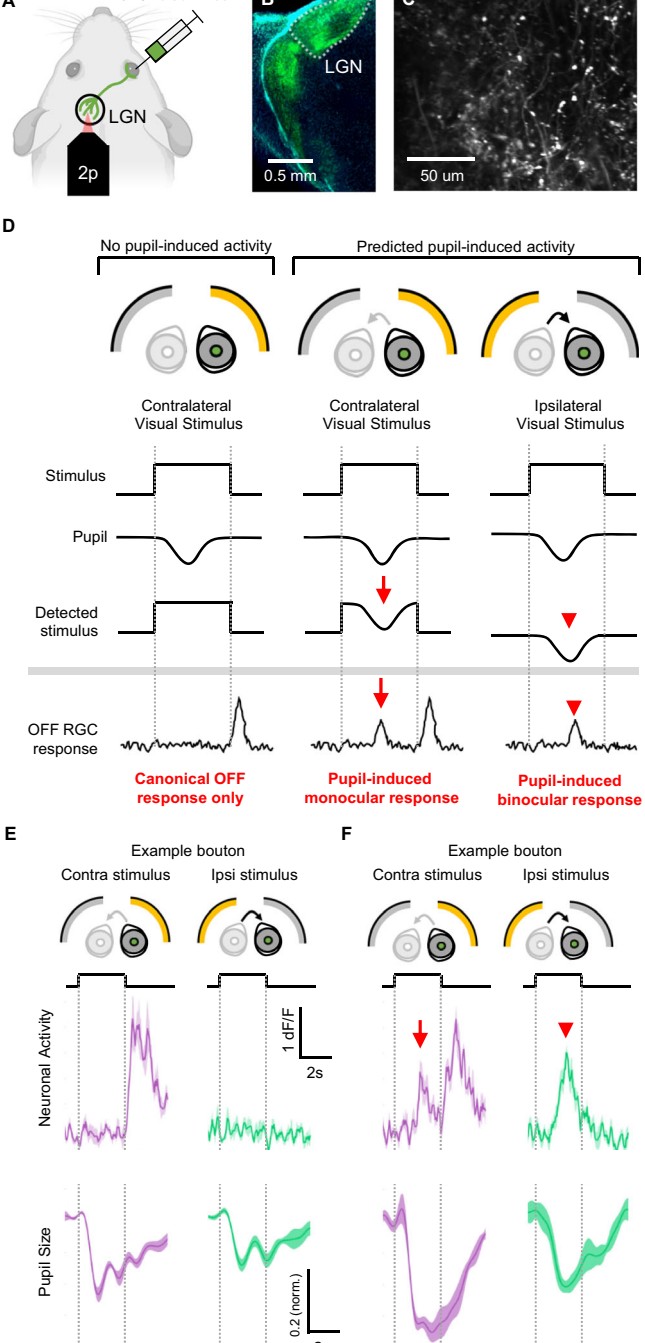

**Fig. 1 | Pupillary constriction's predicted monocular and binocular responses in RGC axons in vivo, compared to canonical responses. A** Schematic of retinal AAV injection to label RGC axons in the dLGN (green) for in vivo 2-photon imaging. **B** RGC axons in the dLGN **(C)**, RGC boutons during 2-photon imaging (representative of N = 12 animals). **D** Schematic of possible OFF cell responses: an OFF cell where pupillary constriction plays no role in the response, it only shows a canonical response at light offset to stimulus presented to the injected eye, from the screen contralateral to the recording site (left), an OFF cell exhibiting a predicted pupil-induced response from contralateral screen stimulus (middle, red arrow), an OFF cell exhibiting a predicted pupil-induced binocular response to stimulating the non-injected eye from ipsilateral screen stimulus (right, red arrowhead). **E** Example RGC activity showing canonical OFF response to contralateral eye stimulation (left, purple) and no binocular response to ipsilateral eye stimulation (right, green), contralateral pupil responses to each stimulus in bottom. Data are presented as mean ± SEM. **F** Example RGC showing a response consistent with predicted pupil-induced monocular response to contralateral eye stimulation (left, purple, red arrow), as well as a binocular response to ipsilateral eye stimulation (right, green, red arrowhead), contralateral pupil responses to each stimulus in bottom. Data are presented as mean ± SEM. All flashes are from a 9.5 Lux baseline to 31 Lux.

To rule out light contamination as a possible source of binocular activity, we inactivated the ipsilateral eye with intravitreal injection of TTX, preventing any action potential firing in RGCs of that injected eye. This inactivation would eliminate any binocular responses but not responses due to light contamination or direct PLR-induced activity. Indeed, we found that the binocular non-canonical OFF responses in OFF cells were entirely eliminated by TTX (Fig. 2L, right trace, Figs. 2M and S6). Importantly, direct PLR-induced activity of the contralateral eye itself was maintained (Fig. 2L, left trace, *n* = 187 boutons, N = 3 animals, Figs. 2M and S6), as expected for a purely monocular response mediated by the contralateral eye.

Altogether, these data suggest that two non-canonical visual responses can be induced by the pupil in OFF RGCs: a monocular and binocular one, with less salient responses in ON cells. In addition, we performed correlation of spontaneous activity to determine how many boutons were from a single axon, and, consistent with previous work, found up to 6.5 boutons per axons, all statistical calculations corrected for this possible confound (Fig. S7, Supplementary Methods).

Furthermore, we show that the change of luminance on the retina due to PLR is sufficient to drive the observed non-canonical responses. As the photoisomerization rate is linearly proportional both to the luminance of the screen and pupil area[8,33], we could modify the luminance of the stimulus to model the effect of luminance change due to PLR on animal during pharmacological maintenance of a fully dilated pupil (with application of atropine). Modeled PLR stimulation in constitutively dilated animals also produced non-canonical RGC responses, comparable to PLR-driven responses in both OFF and ON RGC boutons (Fig. S8, *n* = 887 boutons, N = 3 animals). This finding suggests that change in light delivered to the retina alone can account for PLR-induced non-canonical responses.

### The PLR drives visual responses in all RGC cell types
Since we found that binocular RGC responses occur purely due to pupillary constriction, we can use ipsilateral stimulation to determine how different RGC types respond to PLR. We used the full-field chirp stimulus, previously shown to easily allow for functional categorization of the various RGC types in the retina[34] and presented it either to the contralateral or to the ipsilateral eye. Using responses to contralateral chirp to classify RGCs, we obtained 5 classes of RGCs (Fig. 3A) with axons interspersed across the dLGN (Example field of view, Fig. 3B). Chirp-determined RGC types were: one transient OFF type, two sustained OFF types, one transient ON type, and one sustained ON type (Fig. 3C, types 1–5, respectively *n* = 1492 boutons, N = 3 animals). Stimulation of the ipsilateral eye elicited PLR-driven responses in each

varying between animals, as has also been reported[32]. An animal's ability to have robust pupillary constriction from presented stimulus correlated with the animal exhibiting pupil-induced monocular and binocular responses (Fig. S4). To demonstrate that both direct and binocular non-canonical OFF responses in OFF RGC boutons occur due to pupillary constriction, we pharmacologically blocked the PLR in two complementary ways, one keeping the pupil fully dilated, the other fully constricted. Application of atropine to the contralateral eye kept the pupil fully dilated throughout the visual stimulation and carbachol kept it fully constricted (Fig. S5A–C, *n* = 848 (atropine) 307 (carbachol) boutons, N = 4 (atropine) 3 (carbachol) animals). The subsequent lack of pupillary dynamics successfully abolished both direct and binocular non-canonical OFF responses while keeping the canonical OFF responses intact (Figs. 2I–K and S5D). Similar effects were observed in ON RGC boutons (Fig. S2, *n* = 71 boutons, N = 5 animals).

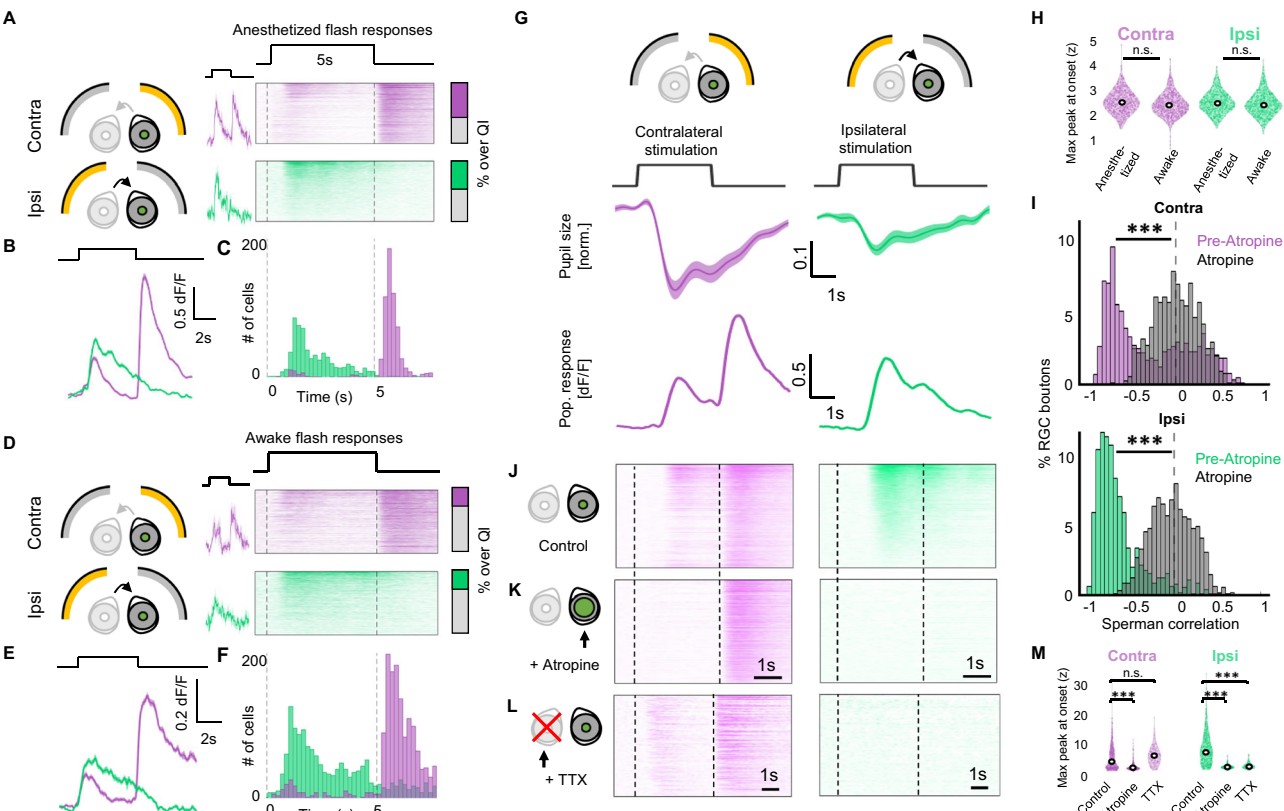

**Fig. 2 | Binocular RGC activity is due to pupillary constriction. A** RGC responses to full-field flash stimulus to the contralateral (top, purple) or ipsilateral eye (bottom, green) in anesthetized animals, for example response (middle, mean ± SD) and heatmap of individual boutons (right). QI = quality index (see "Methods") (*n* = 649 boutons N = 5 animals). Throughout figure, purple indicates contralateral visual stimulation and green ipsilateral stimulation. **B** Population average of heatmaps in A (mean ± SEM). **C** Histogram of responses in (**A**). **D** Same as (**A**), but for awake, freely moving RGC responses (*n* = 709 boutons, N = 4 animals). **E** Population average of heatmaps in (**D**). **F** Histogram of responses in (**D**). **G** Pupillary constriction from full-field stimulus (middle) to either contralateral (left) or ipsilateral (right) stimulation (mean ± SEM), and RGC population responses in the same animals (bottom) (mean ± SEM *n* = 849 boutons, N = 5 animals). **H** Peak non-canonical OFF responses (normalized, RMI) in anesthetized and awake animals to contralateral or ipsilateral stimulation (*n* = 649, N = 5 and *n* = 709, N = 4, respectively). Two-sided two-sample *t*-test. Violin plot with median labeled as white point. **I** Spearman

correlation between RGC activity and pupillary constriction due to contralateral (top) or ipsilateral (bottom) stimulation, before and after atropine. ***p < 0.001 (two-sample two-sided Kolmogorov–Smirnov test) (*n* = 849 boutons, N = 5 animals, *n* = 848 boutons, N = 4 animals, respectively). **J** Heatmap of RGC axons imaged in G. **K** RGC responses of boutons in (**G–I**) after administration of atropine to the contralateral eye, eliminating non-canonical OFF responses due to contralateral stimulation (left) and ipsilateral responses entirely (right). **L** Heatmap of RGC axons after TTX injection into the ipsilateral eye, seeing no change in contralateral activity, including the non-canonical OFF response (left), but completely eliminating ipsilateral responses (right) (*n* = 187 boutons, N = 3 animals). (**M**) Peak non-canonical OFF responses for contralateral (left) or ipsilateral (right) stimulation shown in (**I–K**). ***p < 0.001 (two-sample two-sided *t*-test, multiple comparisons corrected with Bonferroni correction). Violin plot with median labeled as white point.

of the 5 RGC classes (Fig. 3D). In line with observed pupillary dynamics, where the pupil constricts only after at least 0.51 s of continuous stimulation, chirp stimulation elicits a PLR only during flash presentation (Fig. S9 red dotted box, N = 3 animals). Retinal responses during chirp (Fig. 3D red arrowheads) similarly only occurred during prolonged (>0.51 s) full-field flashes and not during fast-changing stimuli (Fig. 3D, red dotted box).

To determine the types of responses driven by the consensual PLR, we next clustered the responding RGC boutons from Fig. 3D based on their response to the chirp stimulus presented to the ipsilateral eye (Fig. 3E). We found two types of PLR-driven responses: a delayed-transient ON response (Fig. 3F top) and a delayed oscillating ON response (Fig. 3F bottom). When further analyzing the temporal profile of PLR-driven responses (Fig. 3G) we can observe that, as expected, response increase in OFF cells and decrease in activity in ON cells occurs during pupillary constriction (between 0.51 and 1.74 s after the onset). Interestingly, some ON RGCs (especially transient ON RGCs), also responded with an increase in activity during rebound pupillary dilation after the PLR peak at 1.74 s (Fig. 3G, F insert zoom).

This suggests that RGC visual responses can also be driven by different types of pupil modulation.

The two types of responses were also scattered across the field of view (example field of view in Fig. 3H). The delayed-transient ON ipsilateral response was found to be present in the OFF RGC types (52.4, 30.6, and 47.2%, for types 1, 2, and 3), while the oscillating response was present in the ON RGC types (13.3 and 40.4% for types 4 and 5) (Fig. 3I). This demonstrates that PLR-driven binocular responses in RGC axons are present in all detected RGC types but qualitatively differ in ON versus OFF boutons, pointing to a global pupil-driven activation of the two channels.

The global effect of the PLR on RGC responses is further apparent when looking at their receptive fields. As the luminance transient due to pupillary constriction affects the entire retina, we could predict that PLR would drive visual responses regardless of the receptive field of the RGCs. We tested this by combining ipsilateral chirp stimulation with receptive field mapping of the contralateral visual field[34]. We observed consensual PLR-driven binocular visual responses in contralateral RGC boutons with receptive fields both in the binocular and

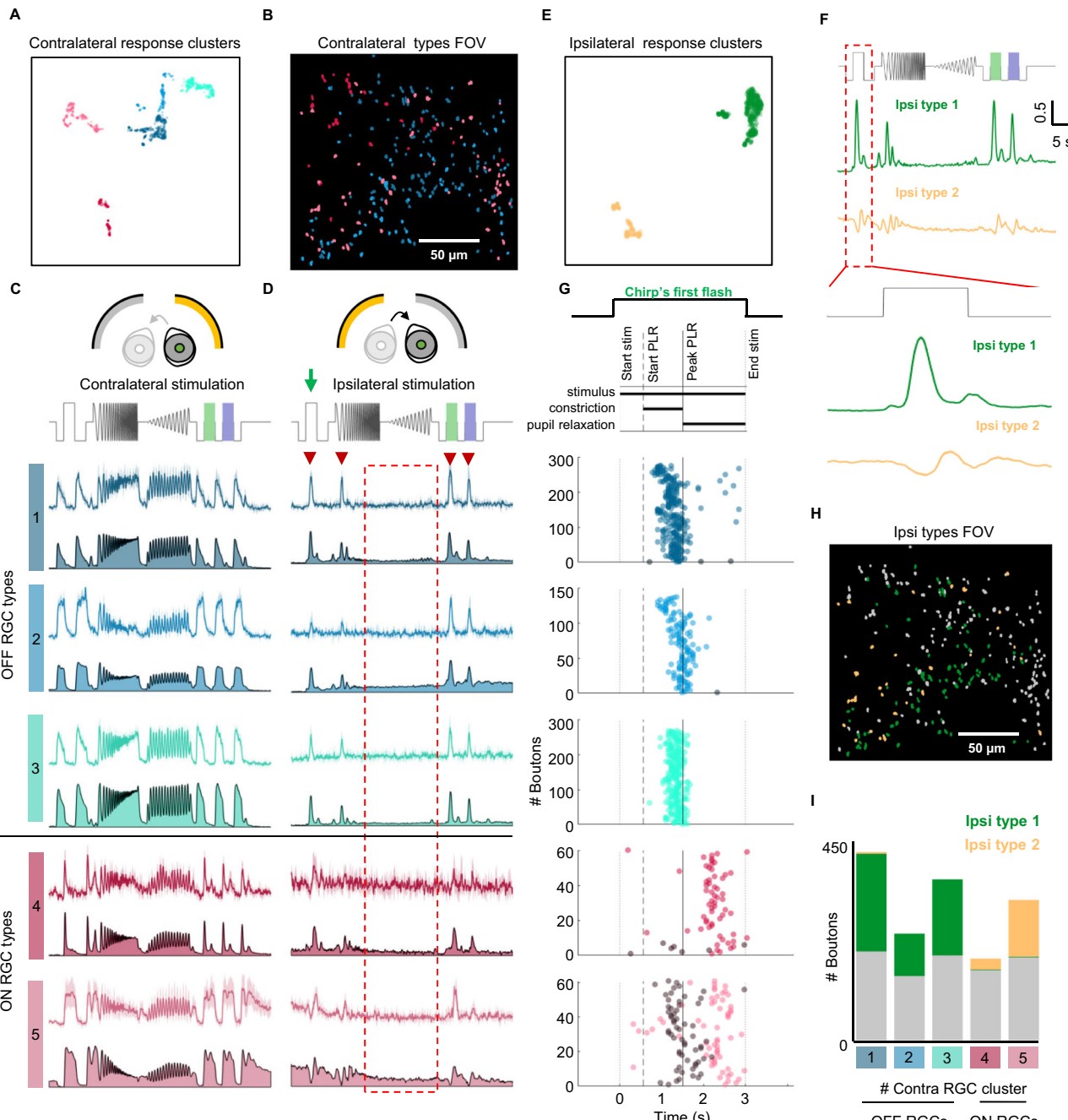

**Fig. 3 | Pupil-induced RGC responses are present across RCG types. A** UMAP of 5 clusters of RGC responses to contralateral full-field chirp stimulus (each with unique color). **B** Example field of view, color coded with cluster identity (representative of N = 3 animals). **C** Responses of RGC boutons to chirp presented to the contralateral eye, showing 3 OFF RGC and 2 ON RGC types (each with unique color as in (**A**)). Top row is example RGC, bottom is population average (n = 1492 boutons, N = 3 animals). **D** Responses of each RGC cluster in C to chirp presentation to the ipsilateral eye, color-coded as in (**A**). Arrowheads = responses consistent with pupil-induced responses. Box = lack of activity in amplitude or frequency ramps. **E** UMAP of 2 clusters of RGC ipsilateral responses to chirp stimulus in D (yellow and green). **F** Population responses of ipsilateral chirp responses: type 1 (middle, green) = a delayed ON response, type 2 (bottom, yellow) = a delayed OFF response

followed by oscillating rebound. **G** Top: PLR phases during 3 s Full field flash (Supplemental Fig. 4). For boutons in each RGC subclass: peak response time (color) and suppressed response time (gray) to ipsilateral stimulation during the Full-Field Flash section of chirp stimulation for each bouton passing response threshold (OFF RGC types: 1: peak = 276, suppressed = 3, 2: peak = 137, suppressed = 1, 3: peak = 268, suppressed = 0. ON RGC types: 4: peak = 60, suppressed = 7, 5: peak = 61, suppressed = 61). Color-coded based on their contralateral response as in (**A**). **H** Same example field of view as 3B, with 2 ipsilateral clusters color-coded based on ipsi cluster obtained in (**D, E**). (representative of N = 3 animals). **I** Prevalence of each ipsilateral response in the 5 contralaterally-determined clusters, with type 1 (green) overwhelmingly in OFF RGCs and type 2 (yellow) in ON RGCs.

monocular part of the visual field (Fig. S10A). Furthermore, there was no difference in the percentage of RGCs responding to the consensual PLR between FOVs with receptive fields located in different parts of the visual field (Fig. S10B, C).

## PLR-driven retinal responses get relayed to the visual cortex

Next, we wanted to show that the PLR-driven visual responses get relayed to the visual cortex. To do so, we injected an AAV expressing GCAMP8f into the dLGN (Fig. 4A). We then recorded the activity of the

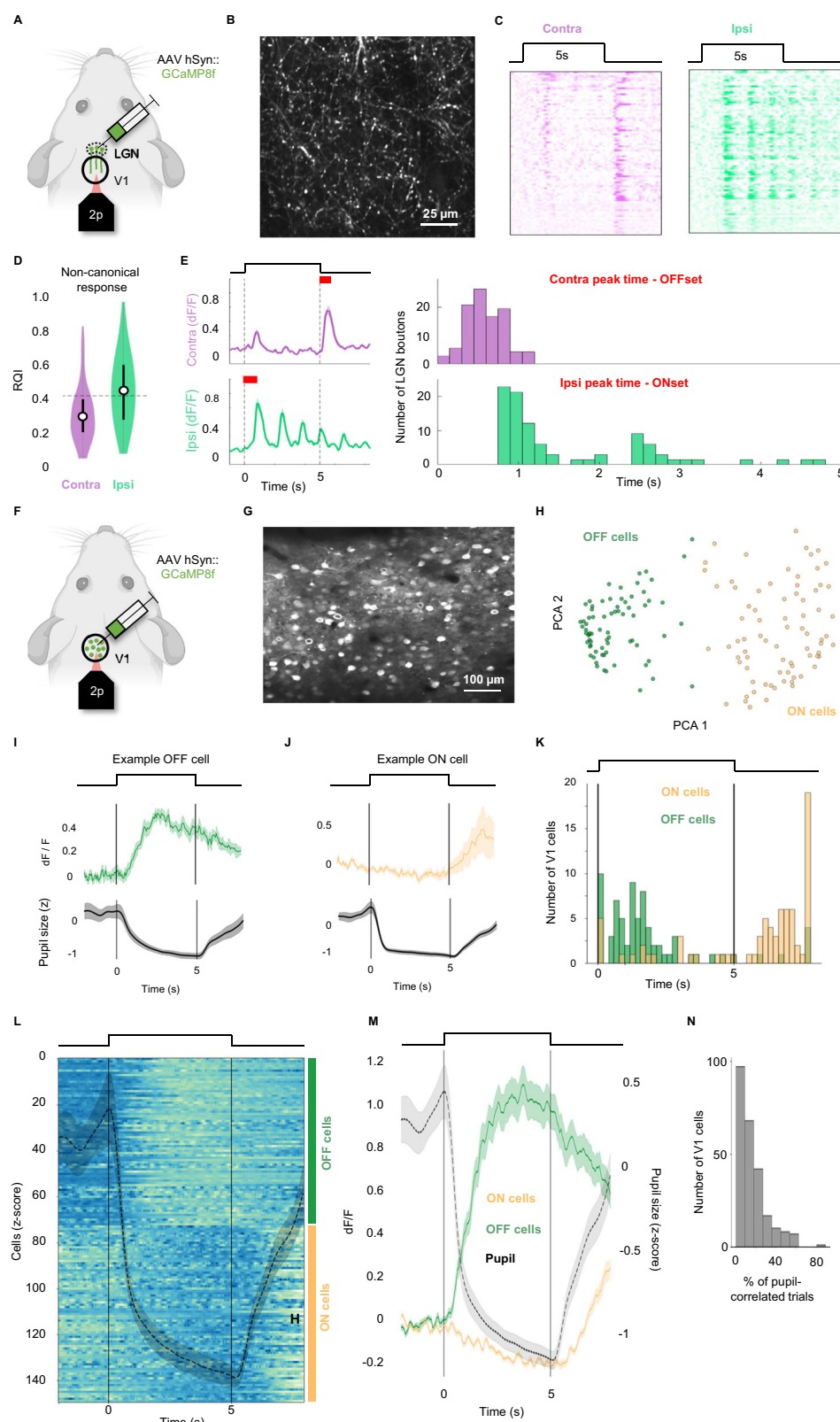

dLGN axon terminals in the primary visual cortex (V1) in response to chirp and full-field flash stimulation of either the contralateral or ipsilateral eye (Fig. 4B).

Unlike RGCs, dLGN neurons are known to receive inputs from both eyes[35–37]. However, previous work reported limited functional binocularity of the dLGN boutons in V1, with only 7% of boutons having strong binocular responses[38,39]. Due to their experimental conditions

(with the opposite eye in the dark), their data would not detect PLR-induced binocularity, similar to our experiments with a dark background (Fig. S3). In contrast, when using a gray background to allow for PLR responses, we show that 56.1% (48 out of 82, N = 3) of dLGN boutons in V1 respond to full field flash presented to either eye (Fig. 4A–E). Furthermore, responses to ipsilateral stimulation have characteristics comparable to those observed in PLR-driven responses

**Fig. 4 | Pupil-induced RGC activity is relayed to the visual cortex. A** Experimental setup. 2-photon imaging of GCaMP8f expressing LGN axon terminals in V1 (green). **B** Example field of view (representative of N = 3 animals). **C** Responses of LGN axons to full field visual stimulation of contralateral (purple) or ipsilateral (green) eye (*n* = 82 boutons, N = 3 animals). **D** Response quality index of responses during stimulation. Contralateral stimulation (left, purple): mean = 0.24, above response threshold (0.3): 18.3%. Ipsilateral stimulation (right, green): mean=0.31, above response threshold (0.3): 56.1%. (*n* = 82 boutons, N = 3 mice). Boxplot values, from left to right: Mean (0.24, 0.31), lower box (0.19, 0.23), higher box (0.29, 0.39), lower whisker (0.12, 0.13), higher whisker (0.42, 0.57), max (0.5, 0.57), min (0.12, 0.13). **E** Response dynamics of LGN responses. Left: Population response of boutons to contralateral (top, purple) and ipsilateral (bottom, green) stimulation. Right: Time of peak response. Right top, Purple: the difference between stimulus offset and maximal response to contralateral stimulation. Right bottom, Green: the difference between stimulus onset and maximal response to ipsilateral stimulation. Red: time

shown in histogram. **F** Experimental setup of layer 2/3 neurons in V1 (green). **G** Example field of view (representative of N = 3 animals). **H** K-means clustering in PCA space of the average responses of V1 cells (*n* = 152 cells, N = 2 animals), 2 clusters: OFF (green) and ON (yellow)-responding. **I, J** top: examples of one OFF (green) and one ON (yellow) cell response. Black, vertical lines = beginning ipsilaterally-presented LED flash. Bottom, gray: trial-averaged pupil area traces (z-score). Solid trace = mean, Shaded areas = SEM. **K** Histogram of the delay onset, for all the cells, color-coded according to the functional cluster (OFF green, ON yellow). **L** Average responses (*n* = 152 cells, N = 2 animals). Black, vertical lines = flash; the black dashed line = mean pupil response ± SEM. Blue is an activity increase, teal decrease. **M** Population response (OFF in green, ON in yellow), plotted against pupil response (gray, same as in (**L**)), all mean ± SEM. **N** Histogram of the percentage of the trial correlated with the pupil, calculated for all the cells recorded (*n* = 250 cells, N = 2 animals).

---

in the RGC boutons. Namely, they occur during the stimulation with a characteristic delay after the onset of the stimulation (Fig. 4E). Additionally, like RGC boutons, dLGN boutons also showed PLR-driven responses only during constant, prolonged stimulation, but not during high frequency, flickering portion of the chirp stimulus (Fig. S11, red arrows in S11D, *n* = 124 boutons, N = 3 animals). Together, binocular response prevalence and characteristics suggest that dLGN boutons in V1 exhibit PLR-driven responses.

To check whether the signal recorded in the RGCs and dLGN boutons actually reached V1 neurons, we performed in vivo calcium imaging recordings of cell bodies in the monocular zone of layer 2/3 of V1 in awake animals (Fig. 4F, G). A constant gray screen was placed in front of the contralateral eye and an LED was used to stimulate the ipsilateral eye, in order to specifically evoke PLR-driven responses. While full-field flash stimuli are notably less able to induce responses in V1 compared to the retina or dLGN, we found a significant population of cells (*n* = 150/250, N = 2) that exhibited either positive or negative strong correlation with the pupil dynamics during the evoked PLR events (Fig. 4H–M). The two populations were clearly separated in the PCA space and showed response dynamics reminiscent of the ones we found in the OFF and ON RGCs populations. Particularly, the OFF responses (Fig. 4I) presented the typical delay consistent with the PLR onset (Fig. 4K) and clearly correlated with the pupil constriction, whereas the ON cells were negatively modulated after the onset of the PLR and correlated with the pupil re-dilation following the PLR event. Overall, these data further support the previous findings in the dLGN boutons, suggesting that the PLR-driven retinal activity is relayed to the visual cortex.

### Binocular facilitation at low contrast from consensual PLR-driven RGC activity

Stimulation of either eye leads to constriction of both pupils and, therefore, to PLR-driven visual responses. Stimulating both eyes at the same time, however, has been shown to enhance pupillary constriction compared to stimulating only one eye[21–23]. We therefore predicted that stronger pupillary constriction following binocular stimulation could lead to potentiation of non-canonical, PLR-induced retinal responses. Since binocular stimulation is known to assist with the detection of low-contrast stimuli, a process known as binocular summation[40], we decided to probe the effect of binocular stimulation on the PLR-driven responses using low-contrast and high-contrast luminance changes.

Binocular summation theoretically requires that the activity driven by both eyes during low contrast stimulation is boosted compared to the activity driven by the dominant eye alone. We asked whether the synergistic interplay between the direct and consensual PLR during stimulation of both eyes could potentiate the PLR-driven retinal responses to low contrast luminance changes (Fig. 5A).

We recorded RGC axonal activity while presenting either high- or low-contrast full-field flashes either to the contralateral eye or both

eyes (for details on light levels chosen, see Figs. S12 and S13A–D). We identified 2 OFF and one ON functional populations (Fig. S12, see "Methods"). For the first OFF RGC type, the canonical response to the stimulus offset remained constant (Fig. 5B, C red arrowhead). However, the PLR-induced component of the responses underwent significant binocular facilitation only during low contrast stimulation (Fig. 5B, C red arrow, *n* = 287 (low contrast), 519 (high contrast) boutons, N = 5 animals). While the amplitude of the effect was stronger in the population reported in Fig. 5, both the OFF populations showed significant facilitation during the low-contrast luminance change stimulus (Fig. S13).

Furthermore, we observed increased contralateral pupillary constriction in response to stimulation of both eyes compared to the contralateral eye alone during low contrast (pupil facilitation index (PFI) = 0.16, p = 0.005) but not high contrast stimulation (PFI = 0.03, p = 0.48) (Fig. 5D top, N = 5 animals). This pupillary constriction correlated with binocular facilitation (red arrow in Fig. 5D bottom). Low contrast stimulation of both eyes led to facilitation in 46.3% of boutons, while only 3.2% were binocularly facilitated during high contrast stimulation (Fig. S13E, F, *n* = 133/287 (low contrast), 17/519 (high contrast) boutons, N = 5 animals). The described binocular facilitation was limited to OFF boutons, with ON boutons exhibiting comparable responses to monocular and binocular stimulation (Fig. S14, *n* = 72 (low contrast), 73 (high contrast) boutons, N = 5 animals).

To confirm that low-contrast enhancement of activity was mediated by the PLR, we tested if this response could be eliminated with atropine or carbachol administration to the contralateral eye. This facilitation was indeed eliminated by both atropine and carbachol (red arrow in Fig. 5E, F), including an elimination of the correlation between RGC activity and the pupil (Fig. 5G). Carbachol induced a pupil-independent change in the shape of canonical OFF responses, as was previously described (Fig. S5C)[41,42].

We further tested if the response to both eyes could be explained by a simple addition of activity driven by the direct and consensual PLR alone (linear summation) or whether the facilitation was more than additive (supralinear summation). In low contrast stimulation, 40.4% of RGC boutons exhibited responses that were more than additive (Fig. 5H left), compared to 0.4% during high contrast stimulation (Fig. 5H right).

These data show that: (1) increased pupillary constriction following binocular stimulation can lead to an increase in the PLR-driven visual responses, and (2) PLR-induced binocular facilitation disproportionately facilitates responses to low contrast, compared to high contrast luminance changes.

### The dynamics of pupillary constriction constraint PLR-driven RGC activity

We proposed that a change in luminance on the retina due to PLR leads to visual responses in RGCs, and that the increased pupillary

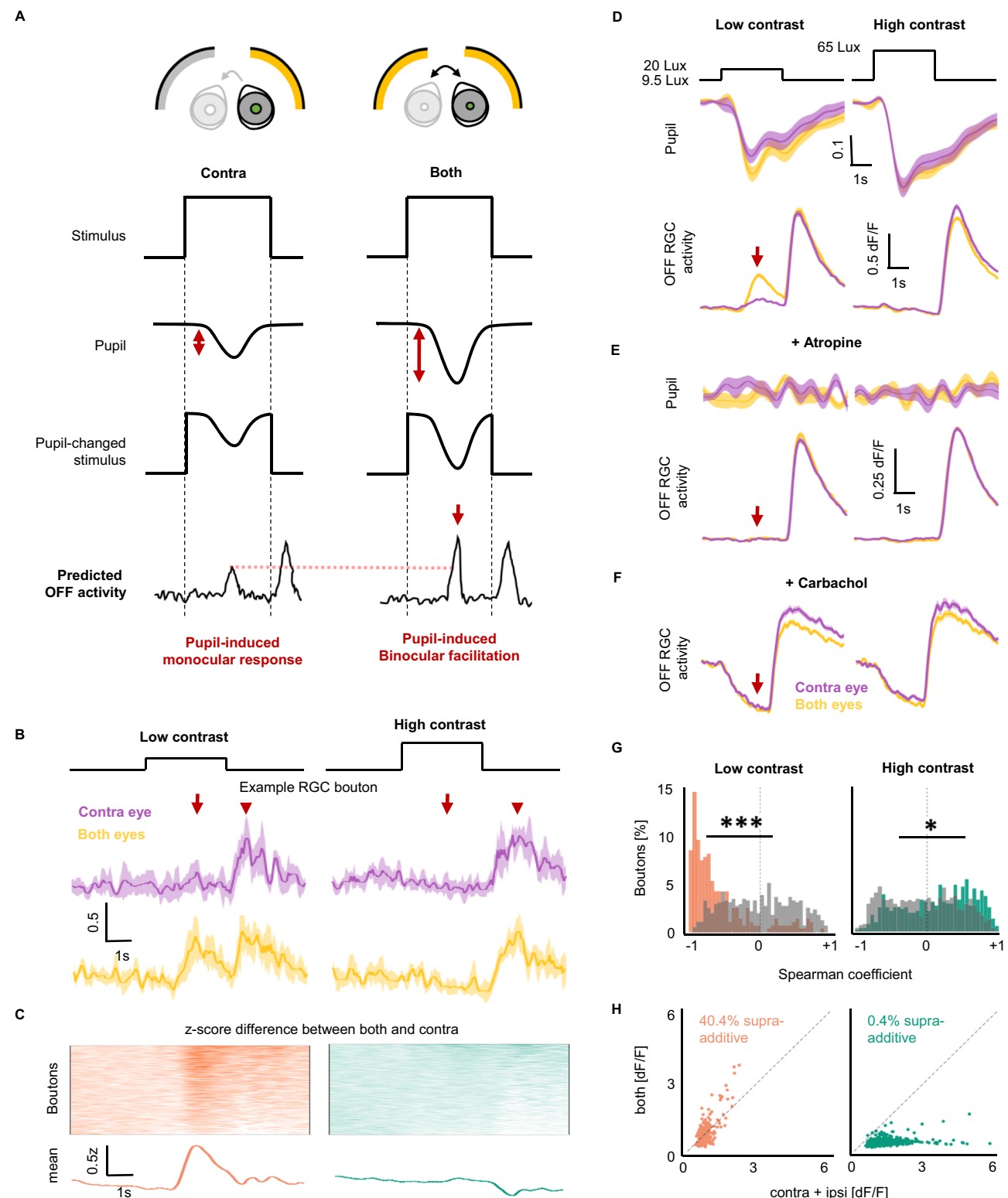

constriction following binocular stimulation leads to low contrast binocular facilitation. However, it is currently predicted that a smaller pupil itself could improve visual detection by reducing blur[13–15]. To decouple the two effects and show that a transient change in luminance leads to binocular facilitation, rather than a smaller final size of the pupil, we first relied on the temporal dynamics of pupillary constriction.

As shown above, the pupil takes 0.51 s to start constricting, with RGCs responding with a ~0.9 s delay after the onset of the stimulation.

Therefore, a sinusoidal full-field stimulus with temporal frequency slower than time to constriction onset (0.55 Hz) would lead to a PLR at each response peak (Fig. 6A left, S15). However, the pupil would not follow a high-frequency stimulus presented at 2 Hz. Rather, during high-frequency stimulus presentation, the pupil would gradually constrict throughout the stimulation (Fig. 6A right)[20]. This way, the 2 Hz contrast ramp would achieve a smaller pupil size, but not the fast luminance transients induced by the 0.55 Hz contrast ramp. Should the pupillary dynamics and not merely the final state of the pupil be

**Fig. 5 | Pupil-induced binocular modulation facilitates low contrast responses. A** Predicted pupil-induced response in RGCs to contralateral stimulation (Left) and both screen stimulation (Right), showing increased pupil constriction (compare double red arrow in Contra to Both conditions), inducing predicted pupil-induced monocular response (Left, reaching dotted red line) and predicted binocular facilitation (Right, arrow) in OFF RGC responses. **B** Example RGC bouton response to low (left) or high (right) contrast stimulation from the contralateral screen (Top, purple) or both screens (Bottom, yellow). Traces are mean ± SD. Arrow = Pupil-induced binocular facilitation (as predicted in (**A**)), Arrowhead = Canonical OFF response. Throughout figure, purple is contralateral eye stimulation, yellow is both eye stimulation. **C** Heatmap (Top) of RGC difference between binocular and monocular responses (z-score), as well as population z-score average (Bottom trace) (*n* = 287 (low contrast, orange), 519 (high contrast, green) boutons, N = 5 animals). **D** Pupil response to contralateral (purple) or both eye (yellow) stimulation (top) compared to population average of RGC activity (bottom). Arrow = predicted binocular facilitation. (*n* = 287 (low contrast), 519 (high contrast) boutons, N = 5 animals). Traces are mean ± SEM. **E** Same as in (**D**), but after atropine administration to the contralateral eye, eliminating binocular facilitation (arrow). (*n* = 218 (low contrast), 776 (high contrast) boutons, N = 4 animals). Traces are mean ± SEM. **F** Same as in (**D**), but after carbachol administration, eliminating binocular facilitation (arrow). (*n* = 133 (low contrast), 307 (high contrast) boutons, N = 3 animals). Traces are mean ± SEM. **G** Spearman correlation between increased pupil contraction and binocular facilitation before (orange, green) and after (gray) atropine for low (Left, orange) and high (Right, green) contrast, with RGC activity to low contrast being substantially correlated to pupil activity. ***p < 0.001, *p = 0.0106 (two-sample two-sided k-s test). **H** Plot of RGC activity during stimulation onset to both eye stimulation (y-axis) against the sum of responses to the individual screens (x-axis) for low contrast (Left, orange) and high contrast (Right, green) (*n* = 287 (low contrast), 519 (high contrast) boutons, N = 5 animals). Low contrast = 9.5 Lux baseline to 20 Lux, high contrast = 9.5 Lux to 65 Lux.

important for binocular facilitation, RGC responses would be facilitated when the pupil is able to track the signal (at 0.55 Hz), but not when it cannot (at 2 Hz). However, if the facilitation was only due to the final smaller pupil size, it should be present in both conditions.

Using a sinusoidal, temporal contrast ramp stimulus, we found that the pupil was indeed able to track the luminance changes if the stimulus was presented at 0.55 Hz (Fig. 6B left). However, when the stimulus was presented at 2 Hz, the pupil constricted gradually throughout the stimulation, not following the luminance change of stimulus (Fig. 6B right). OFF RGC boutons (*n* = 1381 (0.55 Hz), *n* = 648 (2 Hz) boutons, N = 5 (0.55 Hz), 3 (2 Hz) animals) responded to contralaterally presented contrast ramp at both frequencies (Figs. 6C and S16A, right). However, PLR-driven responses were only present when the slower, 0.55 Hz contrast ramp was shown to the ipsilateral eye (Fig. 6D left, S16A, left). When stimulating the ipsilateral eye with a 2 Hz contrast ramp, we observed no responses (Fig. 6D right, S16A). Stimulation of both eyes increased final pupillary constriction compared to monocular stimulation both during the 0.55 Hz contrast ramp and during the 2 Hz contrast ramp in comparable amplitude (Fig. 6E, p = 0.26, see "Methods"). However, only 0.55 Hz stimulus elicited binocular facilitation of the RGC responses (Figs. 6F and S16B), suggesting that a stronger pupillary transient, not a smaller final pupil size, is important for the facilitation of RGC response. This data confirms that fast luminance transients, such as those caused by the PLR, are necessary for pupil-driven visual responses. Therefore, pupil-driven activity from the PLR is not due to any effect of a smaller pupil compared to a bigger one, but a retinal response to a sharp luminance change.

Binocular facilitation of boutons in Fig. 6B–D during 0.55 Hz stimulation was strongest during the early, low-contrast part of the contrast ramp stimulus (mean RMI of 0.23 ± 0.01SEM at 3rd contrast step), with less facilitation present throughout the stimulation (mean RMI 0.1 ± 0.004 SEM at 9th contrast step), consistent with the PLR-induced RGCs responses being preferentially potentiated by low contrast stimuli (Fig. 6G, H, middle, blue). This facilitation was entirely eliminated after the application of atropine (Fig. 6G, H, bottom, gray). Binocular presentation of 2 Hz stimulus failed to facilitate responses (Fig. 6G, H, top, red).

We further examined low- and high-contrast binocular facilitation by comparing the change in RMI in individual boutons during low-contrast steps (peaks 2, 3 and 4) and high-contrast steps (peaks 9, 10, 11) in anesthetized (Fig. 6I) and awake mice (Fig. 6J right, *n* = 213 (0.55 Hz), *n* = 312 (2 Hz) boutons, N = 3 animals, top 50% modulated boutons for anesthetized condition, all responding boutons for the awake condition, for all boutons in anesthetized condition, see Fig. S16C, D). In both anesthetized and awake animals, responses to low contrast stimulation were more binocularly facilitated than responses to high contrast stimulation only during 0.55 Hz contrast

ramp (Figs. 6I–K and S16D) Thus, RGC boutons exhibited binocular facilitation only for stimuli that the pupil constriction could track (0.55 Hz), such binocular facilitation being enriched for low contrast stimulus.

Additionally, we complemented the analysis of the binocular facilitation happening during the 0.55 Hz contrast ramp by characterizing the improvement in the signal-to-noise ratio (SNR). Previous work on human psychophysics found that binocular viewing increases low contrast detection by a factor of 1.4 (or $\sqrt{2}$), and in some cases by a factor of 2, a phenomenon termed binocular summation, which is predicted to occur at the first point of binocularity[21]. Since PLR-driven binocularity on the retina precedes thalamic or cortical binocular integration, we hypothesized that PLR-induced binocular detection could also improve the SNR by a factor of 1.4-2 compared to monocular detection. Interestingly, we found that the average increase in the SNR observed for the low-contrast portion of the stimulus reached value above 1.4 for a significant proportion of the recorded OFF boutons, with some responses reaching 2-fold improvement (Fig. S17, *n* = 518/1381 (1.4), 209/1381 boutons (2), N = 5 animals)[43–45]. This amount of SNR improvement is consistent with the notion of active, supralinear summation of two signals (as also shown in Fig. 5H), also reported in the literature of perceptual binocular summation.

## PLR dynamics are comparable between mice and humans
Since all monocular and binocular PLR-driven responses we found could be attributed to the dynamics of pupillary constriction, we examined if these dynamics are conserved between mice and humans. We found that the dynamics of PLR in response to a full field flash are comparable between mouse and human (Fig. 7A). There was no significant difference between times of constriction onset (mouse = 0.56 s, human = 0.55 s, p = 0.11) (Fig. 7B left) but peak constriction was faster in human (mouse = 1.56 s, human = 1.18 s, p = 0.0070) (Fig. 7B right) Furthermore, in humans, pupillary constriction was also able to track a sinusoidal contrast ramp at a 0.55 Hz (Fig. 7C right), but not at 2 Hz (Fig. 7C left). In both species, at 0.55 Hz the stimulus correlated with pupil dynamics, but at 2 Hz it did not (Fig. 7D, E), meaning that both species share an optimal temporal frequency profile for pupil tracking of sinusoidal signal. This similarity in pupil constriction kinetics suggests that pupil-driven RGC responses may also be present in humans.

## The PLR-driven visual responses generate a perceptual dimming effect in humans
In order to understand whether the PLR-driven visual responses that we found throughout the visual circuit ultimately affects perception, we performed human psychophysics. Volunteers were placed in front of a curved screen and were presented with a sequence of 5-s-long full-field increases in the monitor brightness from a gray baseline. The

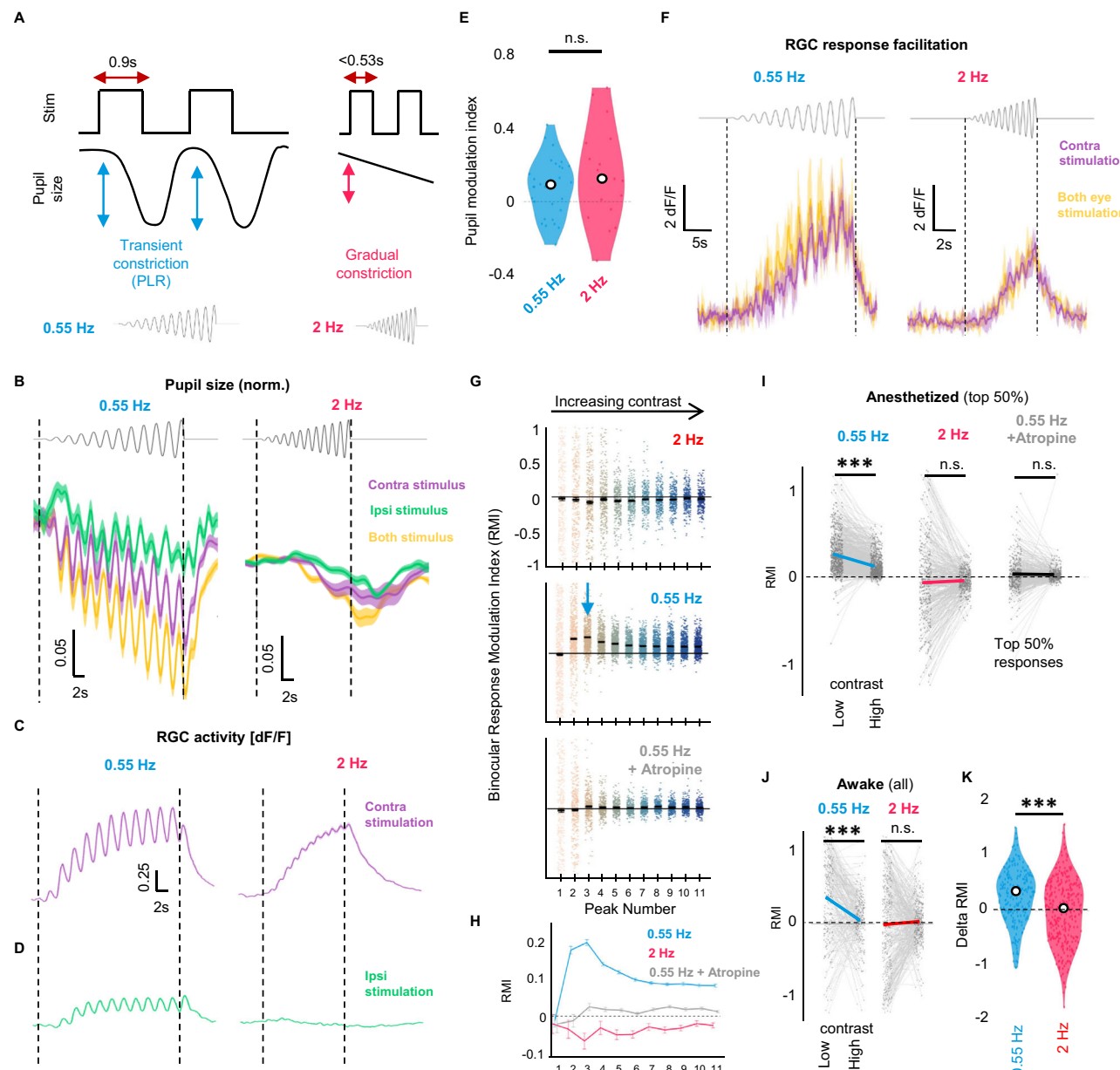

**Fig. 6 | The dynamics of pupillary constriction constrain PLR-driven RGC activity. A** Design of visual stimulus of increasing contrast that allows for the pupil to track each contrast change, by having the flashes presented at 0.55 Hz (left, blue), compared to stimulus of identical properties, but presented at 2 Hz (right, red). Horizontal arrows indicate flash length at each frequency. Throughout the figure, blue indicates 0.55 Hz stimulus, red 2 Hz stimulus. **B** Pupil size of contralateral eye due to contralateral (purple), ipsilateral (green), or both screen (yellow) stimulation at either 0.55 Hz (left) or 2 Hz (right). (N = 5 (0.55 Hz) and 3 (2 Hz) animals. Mean ± SEM. **C, D** Population RGC axon activity from 0.55 Hz (left) or 2 Hz (right) stimulation to either the contralateral (C, purple) or ipsilateral (D, green) eye. (n = 1381 (0.55 Hz), 648 (2 Hz) boutons, N = 9 (0.55 Hz), 5 (2 Hz) animals). **E** Increased pupillary constriction on both eyes compared to the contralateral eye for 0.55 or 2 Hz amplitude ramp. n.s. (two sample *t* test) (N = 5 (0.55 Hz) and N = 3 (2 Hz) animals, 6 repeats). White point = median. **F** Example RGC axon activity when

presented with 0.55 Hz (left) or 2 Hz (right) amplitude ramp, with binocular facilitation (both eye responses (yellow trace) greater than contra eye responses (purple trace)) only observed at 0.55 Hz. Mean ± SD. **G** binocular facilitation (RMI) for top 50% OFF RGCs at each peak of the ramp at 2 Hz (top, red), 0.55 Hz (middle, blue), or 0.55 Hz after atropine administration (bottom, gray). Arrow = peak binocular facilitation. (n = 691 (0.55 Hz), 351 (0.55 Hz + atropine), n = 324 (2 Hz)). Color = peak number. Bars = Mean. **H** Population means for (**G**). bars = SEM. **I, J** Paired binocular facilitation in individual boutons at low contrast (peaks 2, 3, 4) or high contrast (peaks 9, 10, 11) in (**I**) anesthetized, top 50% boutons (n = 691 (0.55 Hz), 351 (0.55 Hz+atropine), 324 (2 Hz) boutons, N = 5 (0.55 Hz), 4(0.55 Hz +atropine), 3(2 Hz)) or (**J**) awake animals, all boutons. ***p < 0.001 (paired two-sided *t* test) (n = 213 (0.55 Hz), 312 (2 Hz) boutons, N = 4 animals. Line = mean. **K** Mean RMI difference at low contrast in (**J**). ***p < 0.001 (two-sample two-sided *t*-test). White point = median.

sequence of flashes was composed of a randomized combination of low or high-amplitude luminance steps comparable to ones we used for our mouse recordings (see methods). Prior to the test, the volunteers were informed that they would be presented with a sequence of luminance steps and that some of them would fluctuate in intensity after flash presentation, with a fluctuation happening at a random

point during the 5-s-long stimulus. In reality, the luminance was kept constant throughout the duration of the stimulus after flash presentation. They were asked to press a button as soon as they detected any change in luminance during the 5-s flash (Fig. 8A). Remarkably, 7 out of 8 subjects reported the perception of a luminance change, namely a dimming, during uniform stimulation (Fig. S18). This

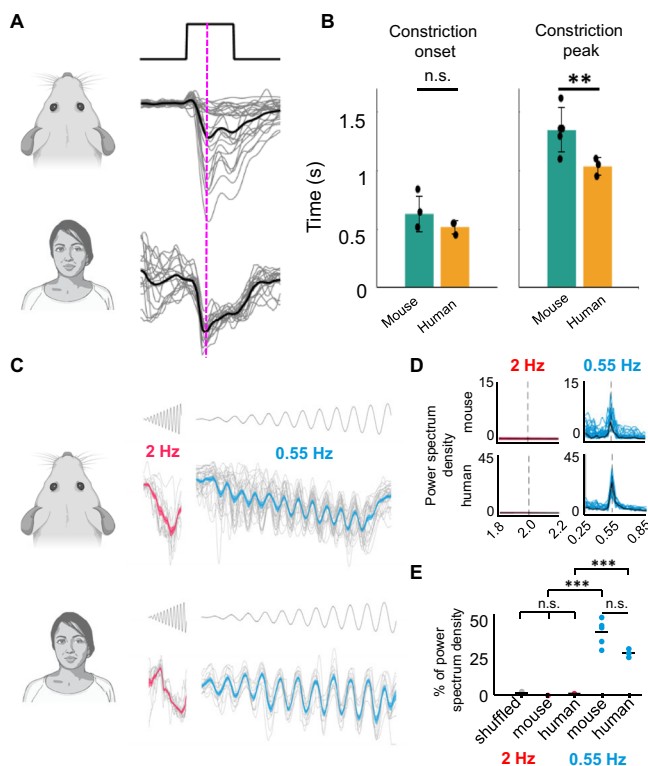

**Fig. 7 | Pupillary kinetics are conserved between the mouse and human.**
**A** Average pupillary response to 3-s full-field flash stimulus in mice (top, N = 5 animals) and human (bottom, N = 3 volunteers), the dotted purple line highlighting peak pupil response coincident with mouse delayed ON RGC responses (see Figs. 1 and 2). **B** Pupillary constriction onset (left) and peak of constriction (right) for data in (**A**) (mean ± SEM) comparing mice (left, green) and human (right, orange). **p = 0.008 (two-sided t-test). **C** Average pupillary response to 2 Hz (left, red) or 0.55 Hz (right, blue) stimulation in mice (top, N = 5 animals) and humans (bottom, N = 3 volunteers). **D** Fast Fourier Transform (FFT) of pupillometry data in (**C**), showing no signal from the 2 Hz stimulus centered on 2 Hz (left, red), but a significant peak from the 0.55 Hz stimulus centered on 0.55 Hz (right, blue). **E** Percent of the entire FFT signal (above 0.25 Hz) at either 2 Hz (red) or 0.55 Hz (blue) for those respective stimuli, for mouse and human. ***p < 0.001 (one-way ANOVA) (N = 3 humans, 5 mice).

perceptual dimming would be consistent with our mouse data if it were seen more frequently with low contrast stimulation than at high contrast. Indeed, across the 8 volunteers this perceptual dimming was reported in 67.2% (43/64) of low contrast stimuli, but only in 26.6% (17/64) of high contrast stimuli (Fig. 8B–D). This was a significantly greater ability to detect perceptual dimming of the stimulus at low contrast than at high (Fig. 8E). Importantly, the time of reported perceptual dimming was 1.2 s (0.13 SEM) for high contrast, 1.1 s (0.06 SEM) for low contrast after the stimulus onset, aligning well with the peak of the PLR event (Figs. 8F and S18). Altogether, these data suggest that the PLR-driven retinal activity we found in mice is likely also present in humans. Furthermore, conscious detection of the PLR-driven activity suggests that this non-canonical retinal modulation plays a hitherto overlooked role in shaping conscious perception.

## Discussion

Here, we demonstrated that pupillary constriction could induce visually-evoked monocular and binocular responses in RGC axons in the dLGN in vivo. These responses were synchronous, globally affecting both the ON and OFF channel, and were relayed to the visual cortex. We found that low contrast changes in luminance induced stronger pupillary constriction when presented to both eyes than

when presented to only one eye. This also translated into stronger PLR-driven retinal responses, showing that pupillary constriction could drive binocular facilitation. Moreover, we showed that this facilitation improved the SNR of the RGC response to binocular stimulation by a factor 1.4–2. Finally, we reported that the kinetics of pupillary constriction driving retinal activity in mice is conserved in humans, and that the PLR is able to affect perception in humans, suggesting that primate vision may also use pupillary constriction for potentiating neuronal responses in the retina. Overall, we propose that pupil-mediated activity can play an active role in shaping conscious vision in daylight conditions.

Binocularity is currently thought to be predominantly a cortical process, with binocular responses defined as a cell responding to stimulus from either eye[25,26]. Canonical binocular responses are enriched in the primary visual cortex, and not prevalent in the dLGN[25,26,35,36,46–48]. In contrast, we found that pupil-driven binocular responses are highly prevalent already at the level of retinal output, both in terms of the percentage of cells exhibiting binocularity (between 27% and 55% of RGCs, depending on the type) and amplitude of the response (over half of a canonical response's maximum dF/F) (Figs. 2 and 3). Previous mouse studies recording RGC activity in vivo did not test for this effect[38,39]. We also found that stimulating both eyes could facilitate RGC activity compared to stimulating one eye alone. This binocular modulation was dramatic for low contrast stimulation but negligible for high contrast stimulus (Fig. 5), was more than additive, and significantly improved the SNR of the produced binocular signal. According to psychophysical studies in human binocular summation, an SNR improvement of 1.4 implies passive addition of two monocular signals, while SNR improvement of 2 would imply an active process[21]. However, such work was performed on faster stimulus that would not be modulated by the pupillary dynamics described here[43–45]. Our observed pupil-mediated non-canonical activity thus likely potentiates other regimes of vision than those that have been previously tested in humans. Overall, a role for the pupil in shaping binocular responses was likely overlooked in both mouse visual circuits and human psychophysics at least in part because the pupil was not considered as a potential driver of retinal activity.

Changes in pupil size are also often used as a readout of arousal in various species[2,49]. Arousal-associated pupil dilation is driven by the activity of the locus coeruleus, but the role of the pupil dilation itself on various neuronal circuits is confounded by the broad noradrenergic modulation the locus coeruleus exhibits through the brain[50]. While retinal axons in both the dLGN and superior colliculus have been previously found to exhibit changes in firing rate correlating with pupillary dilation, the effect was attributed to neuromodulatory influences on retinal axons within the brain, not to visually-evoked retinal activity[1,4,30]. Furthermore, previous work focused on comparing responses at different pupil sizes as a readout of stable arousal states, not triggered by the pupil change itself[4,30]. In light of our finding that pupillary dynamics can drive retinal activity, we speculate that arousal-mediated pupillary dilation could affect visual responses in at least two ways: (1) Pupillary dilation itself could drive visual responses by increasing the luminance on the retina. In this case, the responses would likely be driven mainly in the ON channel—such as via the responses shown in Figs. 3F, G and S1. (2) In the hyper-dilated pupil state, PLR-driven responses would likely get stronger, as the pupillary constriction would lead to a larger luminance transient on the retina.

There are other strong retinal responses that ultimately do not lead to perceptual changes, most famously saccade-induced responses, the retina responding very well to the large motion sweeps induced by a saccadic eye movement, with corollary discharge preventing such responses from affecting the visual cortex. The PLR-induced response we describe, however, is prevalent in the visual cortex (Fig. 4) and can be perceived by humans as perceptual dimming of a sustained full-field flash (Fig. 8). We infer that perceptual dimming

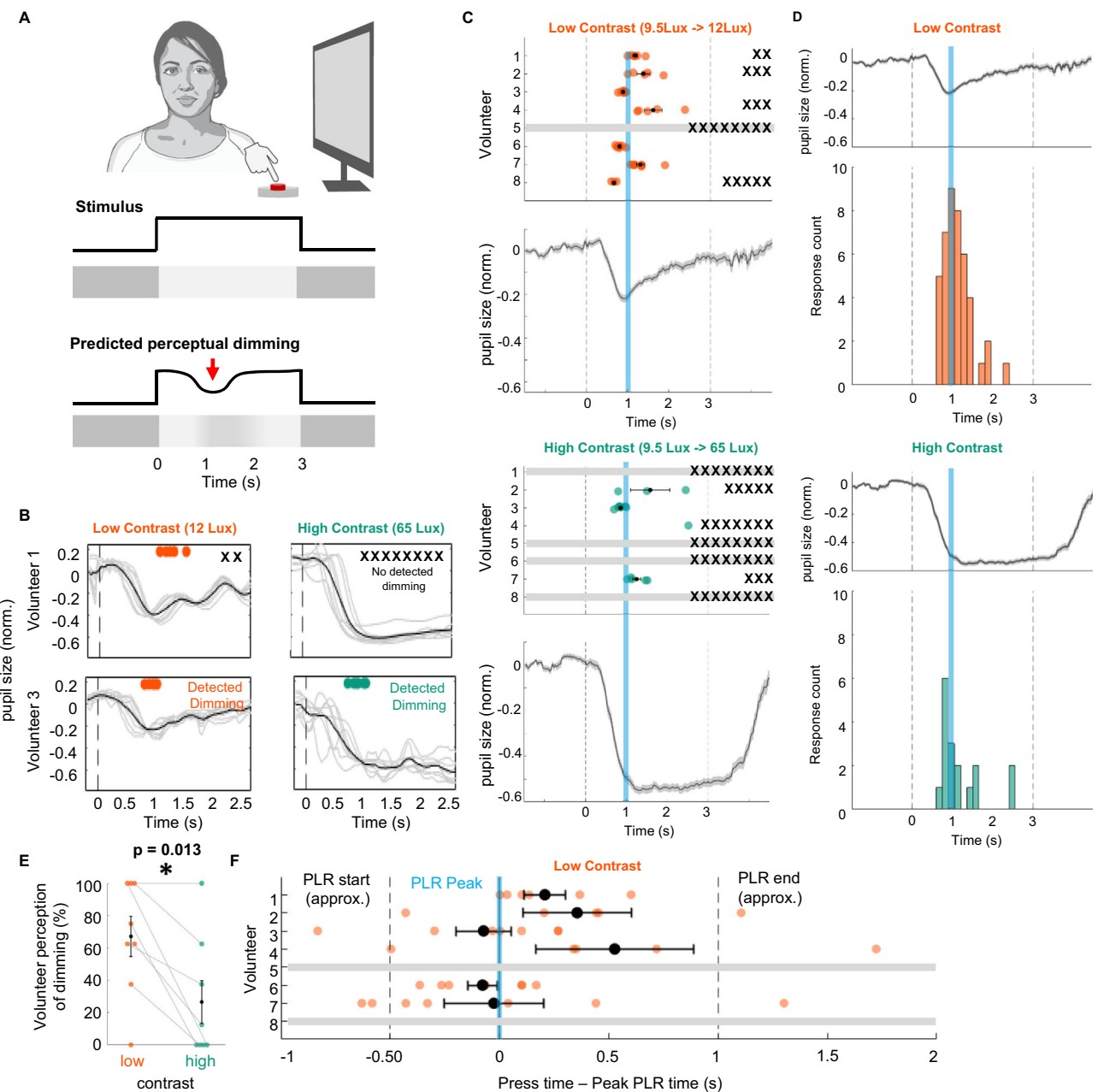

**Fig. 8 | Humans perceive luminance transient coinciding with pupil constriction. A** Setup of human psychophysics experiment. Top: Human subjects (4 male and 4 female) were presented full-field stimulus on computer screen and asked to report perceived change in luminance during a 3-s flash. Middle: Stimulus, a full-field-flash of high contrast (corresponding to LL5 stimulus presented in mice) or low contrast (corresponding to LL1 stimulus presented in mice). Bottom: Prediction of perceptual dimming. We hypothesized that subjects would perceive dimming of the stimulus in time with PLR due to decreased luminance on the retina (red arrow). **B** Examples of 2 subjects reporting perceptual dimming during low-contrast (orange) and high-contrast (green) stimulus. Change in pupil size during stimulus presentation (dark gray: mean, light gray: individual trials). Dots: Time of reported dimming. X: trials with no detected dimming. Throughout figure, orange indicates low contrast stimulus, green high contrast. **C** Perceptual dimming in all volunteers.

Top: Dimming detection during low-contrast stimulation. Time of dimming detection for each of the subjects (colored dots) and number of trials with no detection (black X). The plot is the change in pupil size (line: mean, shaded area ± SEM). Bottom; Same as top, but for high-contrast stimulation. Blue line = 1 s. **D** Detection times shown in C (bottom histograms) aligned to pupil response (top traces). **E** Percentage of trials where dimming was detected during low contrast (orange) compared to high contrast (green) stimulation, $n = 8$, $p = 0.013$ (*), paired two-sided t-test. Black dot is mean across volunteers, bars indicate ± SEM. **F** Difference between each dimming detection and PLR peak time during low-contrast stimulation for all the subjects. PLR start and end (dotted line) are approximated from mean PLR shown in (**C**). ($n = 40$ detection events, N = 6 humans).

found in human volunteers is due to the PLR for two reasons. The first is that the kinetics of this dimming is very delayed, within the range of 1 s after stimulus onset, and thus consistent with the PLR-induced retinal response (Figs. 2, 6 and 7), and not with other well-known rapid, canonical visual processes. Secondly, volunteers detected perceptual dimming better at low contrast than high contrast, despite the higher

contrast eliciting a greater PLR and most canonical visual processes in the human being more robust under high contrast. This is consistent with our mouse recordings, where low contrast stimulation was better at eliciting responses than high contrast stimulus, also despite higher contrast eliciting a greater PLR. Altogether, these links strongly suggest that human perceptual dimming is due to PLR-induced retinal

activity, but future pharmacological trials in humans would further cement the causal link between perceptual dimming and the role of the pupil inducing non-canonical responses.

A key aspect of pupillary dynamics that emerges from our data is how it imposes global, highly coordinated activity onto retinal output. This is a qualitatively different mode of information transfer compared to canonical RGC activity, where the asynchronous activity of different neurons is used to encode the local differences in the highly hetero-geneous information content of the visual scene. Global, synchronous PLR-driven modulation could be used as a pathway for relaying ambient luminance information to the primary visual cortex and higher visual areas. We speculate this strong signal could assist some form of global activity normalization throughout the visual system, helping in the detection of low contrast stimuli under a particular retinal adaptation regime. This could be useful when the average luminance on the retina changes, for example, when a shadow falls on a light source (a leaf above moving to block sunlight). Interestingly, it could also help normalize visual information during non-PLR pupil modulation, such as arousal, when the luminance changes solely due to the change in pupil size, and not due to a change in the environment.

Overall, our findings suggest that instead of being a static shutter that can be understood as having an open or closed state, pupillary dynamics plays a key, nuanced role in shaping conscious vision.

## Methods
### Experimental model and subject details
All the experiments were carried out in accordance with European Union Directive 2010/63/EU and under the approval of the EMBL Animal Use Committee and the Italian Ministry of Health License 23-004_RM_SR. C57BL/6-J (males and females) mice aged between 2 and 6 months were used in the study. Animals were housed using a 12:12-h light/dark cycle, with lights on from 7:00 AM to 7:00 PM, housed at temperatures between 20 and 22 °C, and with relative humidity ranging from 40 to 60%. Source of C57BL/6 animals are internal breeding colonies. The breeders are purchased from Charles River and colony is refreshed every 2 years to avoid a genetic drift.

### Anesthetics
For the general anesthesia induced during both the surgical procedures and the 2-photon imaging sessions, we used intraperitoneal (IP) injections of FMM. An injectable solution of fentanyl (Fentadon, Dechra, 50 µg/ml, 0.05 mg/kg BW), medetomidine (Domitor, Orion Pharma, 1 mg/ml, 0.5 mg/kg BW) and midazolam (Buccolam, Lesvi, 7.5 mg, 5 mg/kg BW) was prepared and diluted in 0.9% saline solution. The solution was then filtered using a 0.1 µm antibacterial filter (HUM) and kept in a container with restricted access at room temperature until use.

### Retinal injections
To label RGC axons, 1 µl of chimeric AAV1.2-ProA5-GCaMP6s ($4.4 \times 10^{13}$ vg, produced at EMBL RomeGEVF) was injected intravitreally in the right eye of 4–6 weeks old mice, using a Hamilton syringe (Merck, cat. HAM7634-01-1EA) with a 23 s G needle (Analytics, cat. no. 7762-05). Mice were first anesthetized using FMM solution (10 µl/g bw, formulation as described above) and head-fixed in a stereotaxic frame (WPI, cat. 505371), while keeping the body temperature to 37 °C using a heating pad (Harvard Apparatus, cat. 55-7023). The sclera of the right eye was pierced using a 30-gauge needle on the dorso-temporal side, and the Hamilton needle was inserted and gently pushed on the retina. Virus was delivered at ~30 nl/s and the Hamilton syringe was extracted after 10 min from the injection. After the injection, 10% buprenorphine (Bupaq, Livisto) was administered subcutaneously, and mice were allowed to recover. GCaMP expression was confirmed histologically after 4 weeks.

### Brain injections
To label dLGN neurons, 200 nl of chimeric AAV1.2-hSyn-GCaMP8f ($2.1 \times 10^{14}$ vg, 1:10 dilution); constructed from pGP-AAV-syn-jGCaMP8f-WPRE (Addgene plasmid # 162376) the virus was injected into the left dLGN. Mice were first anesthetized using FMM (10 µl/g bw, formulation as described above) and head-fixed in a stereotaxic frame, while keeping the body temperature to 37 °C using a heating pad. The hairs were removed and the skin cleaned with iodopovidone (betadine, Pharmavola, cat. 023907292). After performing a single incision with a scalpel, the skull was exposed, cleaned with 0.9% saline and aligned. A small hole was drilled using a dental drill (Foredom, cat. 1070) with steel a 0.25 diameter drill bit (Jota, cat. 515) above the dLGN of the left hemisphere, at −2.2 antero-posterior and −2.07 medio-lateral from Bregma. The virus was delivered 3 mm below the brain surface through a pulled glass needle using an automatic injector (Nanoliter 2020 injector, WPI cat. 300704), at a flow rate of 10 nl/s. Ten minutes after the injection, the glass needle was extracted and the skin was sutured. For V1 injections, 200 nl of AAV9-hSyn-Soma-GCaMP8f (addgene 169258) were injected in 2 locations (−4.5 AP, −2 ML and −3.9 AP, −2.5 ML) 600 µm from the dura, in the right hemisphere. Animals were then injected subcutaneously with buprenorphine and allowed to recover.

### Perfusion and sectioning
Animals were deeply anaesthetized using 2.5% Avertin (Sigma Aldrich cat. T48402) injected intraperitoneally, 200 µL/animal and trans cardiac perfusion was performed, first with 1x PBS (Invitrogen cat. 10010023), and then with 4% PFA (Sigma, cat. 457608). The brain and the eyes were collected and post-fixed in 4% PFA overnight at 4 °C. Brains were washed 3 times (2–5 min each) in PBS before embedding them in 2% low melting agarose (Sigma, cat. A9539) and sectioning 100 µm thick coronal sections using a vibratome (Leica Microsystems VT1000s). The slices were collected in 1x PBS and kept at 4 °C until further processing. We dissected the eyes to retrieve the retinas in 1x PBS and flattened them by making cuts in each of the cardinal directions. We stored them in 1x PBS at 4 °C.

### Immunohistochemistry
For antibody staining, we first blocked the brain sections or retinas in PBS with 0.3% Triton (Sigma-Aldrich, cat. T8787) and 10% normal goat serum blocking solution (Vector Laboratories, cat. S-1000-20) for 1 h at room temperature. The sections were then moved into a primary antibody solution containing 0.3% Triton, 3% goat serum, and the chicken-anti- GFP primary polyclonal antibody diluted 1:1000 (Abcam, cat. ab13970). We incubated the brain sections for 1–5 days at 4 °C before removing the antibodies and washing 3 times for 15 min in 0.3% Triton PBS. Secondary antibodies were added (goat anti chicken Alexa Fluor488, Abcam, cat. num. ab150169) diluted 1:500 in PBS 0.3% Triton, 3% goat serum, and incubated for 2 h in the dark at room temperature. Sections were then washed three times in PBS. We mounted the brain sections or retinas on the slides and covered them in mounting media (Thermo Fischer Scientific, cat. P36930) with DAPI (Thermo Fisher Scientific, cat. D1306, diluted 1:1000), before covering with coverslips. The next day, we secured the edges with nail polish. The images were acquired using a Leica Thunder Imager DMi8 microscope.

### dLGN cranial window implantation
After 4 weeks from the retinal injection, a cranial window was opened and a cannula with a coverslip was placed on the surface of the dLGN contralateral to the injected eye, to allow for 2-photon recordings of the axonal terminals of the labeled RGCs. Mice were anesthetized with FMM (10 µl/g bw) and placed on a heating pad set to 37 °C on a stereotaxic apparatus (as described above). After the hairs were removed and the skin cleaned with iodopovidone (betadine), the skull was

exposed and aligned. A ~3.2 mm diameter craniotomy was drilled on the left hemisphere (contralateral to the injected eye), at −1.8 antero-posterior and −2.15 medio-lateral from Bregma, tangential to the skull surface. The brain was kept hydrated using a cortex buffer (125 mM NaCl, 5 mM KCl, 10 mM glucose, 10 mM HEPES, 2 mM CaCl2, 2 mM MgSO4 in ddH$_2$O). The underlying cortical and hippocampal tissue was carefully aspirated using a vacuum pump, exposing the surface of the thalamus. Care was taken to ensure the thalamic surface and the optic tract were not damaged. Afterwards, a 3 mm diameter stainless steel cannula (custom made), previously glued to a 3 mm coverslip using cyanoacrylate glue, was gently lowered into the craniotomy. The surface of the coverslip was slightly pushed onto the surface of the thalamus and then the outer edge was glued to the skull using tissue adhesive (Vetbond, Ubuy, cat. 14639582) and cyanoacrylate glue (Pattex ultra gel, Distrelec, cat. 110-41-180). After cannula insertion, the remaining exposed portion of the skull was covered with cyanoacrylate glue in order to prevent tissue damage, and a titanium head plate was placed ensuring that the craniotomy was correctly centered. The head plate was further fixed to the skull with dental cement (Paladur, Dentag, cat. DL4621578 and cat. DL6421317) to ensure permanent seal. Carprofen (50 mg/ml) (Rimadyl, Zoetis, cat. 10000319) was administered subcutaneously after the surgery, and the mice recovered for 5–7 days.

### V1 cranial window implantation

After 5–8 weeks from the brain injection, a cranial window was opened above the surface of V1 to allow 2-photon recordings of V1-projecting dLGN axons or V1 cell bodies. Mice were anesthetized with FMM (10 μl/g bw) and placed on a heating pad set to 37 °C on a stereotaxic apparatus. After the hairs were removed and the skin cleaned with iodopovidone (betadine), the skull was exposed and aligned. A 4 mm diameter craniotomy was drilled above the left visual cortex (ipsilateral to the injected dLGN) at −4.5 antero-posterior and −2.4 medio-lateral from bregma for dLGN axon imaging, or above the right visual cortex (−4.5 AP, −2 ML) for V1 cell body imaging. A glass coverslip was gently pushed on the surface of the cortex, and then glued to the skull using cyanoacrylate glue. The head plate was then positioned and fixed using dental cement. At the end of the surgery, mice were injected subcutaneously with 10% buprenorphine, and allowed to recover for 5–7 days.

### 2-photon imaging

2-photon calcium imaging was performed using a resonant scanning 2-photon microscope (Neurolabware, custom designed) with a 16×, 0.80 NA, 3.0 mm WD water-dipping objective (Nikon, cat. MRP07220). GCaMP molecules were excited using a Ti:Sapphire laser (Coherent, Chameleon Vision II, 80 MHz) at 980 nm and images were collected using Scanbox software (Neurolabware) at 15.5 frames/s 655 × 510 pixels/frame. The recordings were performed at 4.8× digital zoom for the RGCs axon imaging (~280 × 260 μm FOV size, between 50 and 100 μm below the optic tract) and at 8× for dLGN neurons axon imaging (FOV size ~140 × 130 μm, around 100 μm from the brain surface). To avoid light leaking from the screen into the PMT during visual stimulation, the objective and the space around the head plate was carefully shielded using black fabric (Thorlabs, cat. BK5) and tape (Thorlabs, cat. T137-1.0). For the animals where multiple locations were recorded during RGC axon imaging, the FOVs were selected to be at least 30 μm spaced in the z direction. For recordings on anesthetized animals, mice were injected with FMM at a dose of 5 μl/g bw and a thin layer of ophthalmic ointment (VitaPos, cat. B00WG11SD4) was applied prior to recording. Body temperature was kept at 37 °C by placing the animal on a heating pad. For awake recordings, mice were trained to run on a circular treadmill (custom-made) while head fixed until they were comfortable and not showing signs of excessive stress (10 days, 2 h training per day).

### Visual stimulation

For all the experiments, visual stimulation was provided dichoptically using two flexible 6-in. OLED screens (2880 × 1440, 60 Hz refresh rate, Wisecoco, model TOP060S02K), curved on 3D-printed arcs. Screens were compensated to make their output linear by inverting the gamma correction function. Screens were placed at 10 cm from the respective eye, spanning 70° × 30° visual degrees. To avoid cross-stimulation between the two screens and ensure true separation of the stimuli presented, a black separator was shaped to fit the silhouette of the mouse head and placed between the two screens, isolating each eye from the stimuli presented to the other. Control experiments using TTX on the non-injected eye further confirmed the reliability of the stimuli separation. Animals were adapted to the gray screen for at least 5 min.

To characterize the effect of pupillary constriction on the neural activity of RGC projecting to the dLGN, or on the dLGN neurons projecting to V1, three different full-field stimulation protocols were used. Each stimulus in the different protocols was presented to either the contralateral, the ipsilateral or both eyes for 6 times in a randomized order, for a total number of 18 trials. During the 10-s-long inter-trial pauses, the screen was kept on a constant luminance level. That was crucial for the binocular responses to occur, as the baseline luminance allowed for the consensual PLR to generate a (negative) luminance transient able to evoke a neural response. Such baseline luminance was set to 9.5 LUX. The stimulation protocols were synced with the 2-photon recordings by sending a TTL signal to the acquisition hardware through a digitally controlled arduino. All the stimuli were designed and presented using the Python library PsychoPy[51].

### Full-field flash

The full field flash stimuli consisted of a simple full-field positive luminance step of either 3 or 5 s duration, rising from the baseline luminance. Two different luminance levels were chosen for generating a "LOW" (20 LUX, 20% Michaelson contrast) or "HIGH" (65 LUX, 43% Michelson contrast) contrast stimulus. Additionally, we also probed the pupillary and neural responses at luminance levels of 12.3 LUX, 31 LUX, and 44 LUX, in order to have a more granular estimate of the effect of the luminance step on the binocular responses (Fig. S13D).

### Full-field contrast ramp

The last stimulation protocol consisted in presenting 12 cycles of a linearly ramping sinusoidal stimulus. The stimulus started from the baseline luminance and oscillated around that, with the amplitude of the oscillations linearly increasing with the number of cycles, ranging from minimal to maximal contrast. The minimal and maximal luminance of the stimulus was set, respectively, to 0.15 and 34 LUX. The contrast ramp stimulus was presented at either 0.55 or 2 Hz and was used for assessing the impact of the stimulus frequency on the PLR dynamics and thus on the PLR-driven binocular responses.

### Full-field chirp

The chirp stimulation was designed as previously described [26] and was presented with the main purpose of maximizing the identification of functional clusters within the recorded neural populations. It was obtained by concatenating 3 s long full-field flashes (white, green and blue), a swept-frequency (chirp) from 0.5 to 8 Hz and 12 cycles of a 2 Hz sinusoidal contrast ramp (same as described above), ranging from minimal to maximal contrast. The minimal and maximal luminance of the stimulus was set, respectively, to 0.15 and 34 LUX.

### Pupil constriction modeling

To model the effect of luminance change due to pupil constriction, we first calculated the change in pupil area of the contralateral pupil

during contralateral, ipsilateral, and binocular stimulation in animals that had consistent PLR throughout full field flash stimulation (animals 1, 2, and 3 in Fig. S8). We changed the luminance of each frame of full field flash stimulus in accordance with the percentage change in average pupil size during the time course of the stimulus. We additionally corrected the luminance of the entire stimulation for the change in baseline pupil size after the application of atropine. In the atropine control stimulation, we corrected the luminance of Full Field Flash stimulation only for the change in pupil area due to atropine (dimming the entire stimulus, because a larger pupil allows more light to enter the retina).

## LED full-field stimulation

In order to probe the role of background stimulation on PLR-driven responses (Fig. S3), we designed UV-LED-illuminated domes 1 cm in diameter that fit over each eye, allowing for separate illumination of either eye (Custom-built semi-transparent plastic domes, attached to LED UV (405 nm) source, Thorlabs Cat. num. M405L4). We presented 6 5-s long flashes to either both eyes, the contralateral eye, or the ipsilateral eye. In the "no background" condition, the eye that was not stimulated remained in darkness, while in the "background" condition, UV-LED was constantly illuminating the non-stimulated eye (0.6 μW). During stimulation, UV was set to 1.5 μW in both cases.

## Pupil tracking

During 2-photon recordings, both eyes were recorded using two cameras equipped with a NIR-enhanced CMOS sensor (Basler acA1300-60gmNIR), a 100 mm f/2.8 lens (MVL100M23, Thorlabs) and a 850 nm bandpass filter (Thorlabs, cat. FBH850-40) placed between the sensor and the objective lens. The eyes were illuminated using infrared lamps, and short-pass dichroic mirrors (Edmund Optics, cat. 69-219) were placed in front of each eye for redirecting the NIR light to the cameras, positioned behind the animal. The acquisition of each frame of the cameras was synchronized with the acquisition of the frames of the microscope, aligning the eye videos with the 2-photon recording. The pupil was segmented using MEYE (ref. 52) and analyzed with a custom Python code. The pupil area was calculated by fitting an ellipse to the mask predicted by the algorithm, and the resulting time series low-pass filtered in order to only keep the pupil fluctuations compatible with the slow dynamics typical of the PLR.

## Atropine application

In order to perturb the PLR-mediated binocular responses, we used 0.1% atropine (Atropina LUX 10 mg/ml, Allergan, purchased at pharmacy) to block the PLR on the contralateral eye during 2-photon recordings under anesthesia. We first performed control recordings of the RGC activity in response to a full-field flash stimulation. After 4–5 days, the responses to the same stimulus, in approximately the same FOVs, were recorded after the instillation of 1 drop (~30 μl) of atropine into the contralateral eye once the pupil was completely dilated and no PLR was visible (5–6 min after administration).

## TTX injection

To confirm the reliability of our dichoptic stimulation system and be sure that no light was leaking from one side of the separator to the other, we recorded the activity of RGC boutons in anesthetized mice before and after the intravitreal injection of TTX. Animals were anesthetized with FMM and then the RGC activity was recorded in response to a full-field flash stimulation. After a control recording, animals were injected intravitreally with 1 μl of 1 mM TTX (Tocris, cat. 1069) using a Hamilton syringe with a 23 s G needle. After ~10 min from the injection, animals were placed back under the microscope and the RGC activity in the same field of view was recorded in response to the same full-field stimulus.

## Bouton mask identification and time course extraction

All the image pre-processing, bouton mask identification, and neuropil estimation were done using suite2p (39). The raw videos were first visually inspected, discarding all the recordings where drift in the z-axis was detected. The videos that passed this step were then registered for correcting for x-y motions along the image plane. After registration, the bouton masks were extracted using Cellpose (40) on the enhanced mean image calculated by suite2p (with model = "cyto" and diameter = 5). The automatic segmentation was then manually curated and only the predicted ROIs with correct shape and size were kept. The downstream analysis was performed using custom pipelines (Matlab and Python). First, the neuropil signal was scaled by 0.9 and subtracted from the whole raw trace $F_{RAW}$. The resulting corrected trace $F_{CORR}$ was smoothed with a moving mean window of 4 frames and the activity windows of each trial were extracted using the synchronization file generated during the recording. In order to compensate for possible shift in the baseline fluorescence due to photobleaching, we independently performed $\Delta F/F$ normalization on each individual trial by using as baseline $F_0$ the mean of the activity calculated during the 2 s before the stimulus onset. The extracted traces were further selected according to a response quality index RQI calculated as described previously[26]:

$$RQI = \frac{Var[\langle C \rangle_r]_t}{\langle Var[C]_t \rangle_r} \qquad (1)$$

where $C$ is the T by R response matrix (time samples by number of trials) and $\langle \rangle_x$ and $Var[]_x$ denote the mean and variance across the indicated dimension, respectively. The RQI gives a quantification of how strong and how conserved across trials the responses are. If the mean of the trials is perfectly representative of each individual trial (i.e., all trials are exactly the same) RQI is equal to 1. For all the recordings, the threshold for the RQI was set between 0.3 and 0.45, according to the overall level of GCaMP expression in each of the FOVs, which could impact the strength of the recorded signal. The ROIs which did not pass the RQI threshold were discarded and not considered for the subsequent analysis.

## Axon identification

For having an estimate of how many boutons might belong to the same axon, we performed a noise correlation analysis using spontaneous activity recorded during 20 min in anesthetized animals placed in front of a gray screen. Since FMM can induce slow frequency oscillations, which can drive rhythmic pupillary constriction and in turn driving RGC activity, a single drop of atropine was instilled in the contralateral eye before starting the recording. The correlation analysis was carried out exactly as described by Liang et al.[31]. Briefly: activity traces were z-scored, and activity windows for each trace were selected around the time points where the signal exceeded 3.5 standard deviations above the mean (±700 ms). For each bouton, all the activity windows were concatenated and then cross-correlated with the concatenated activity windows of all the other boutons, obtaining a normalized correlation matrix. After thresholding the correlation matrix at 0.7, the pairwise cosine distance between the correlation vectors was computed and used as distance matrix for the subsequent hierarchical clustering. The dendrogram was cut at 0.85, yielding a maximum number of boutons per cluster around 6.5 in the 3 FOVs tested. This value was then used for correcting for multiple comparisons all the p-values of the statistical tests carried out throughout the paper, by applying Bonferroni correction.

## Response modulation index

To quantify the amplitude of binocular modulation for a stimulus S, we compared the responses elicited by the stimulation of the contralateral

eye (*CONTRA* trials) with the one elicited by the simultaneous stimulation of both eyes (*BOTH* trials). Such quantification was done by calculating the response modulation index RMI, defined as

$$RMI(S) = \frac{AUC(R_{BOTH}) - AUC(R_{CONTRA})}{AUC(R_{BOTH}) + AUC(R_{CONTRA})} \quad (2)$$

Where, for stimulus *S*, AUC(*R_{BOTH}*) and AUC(*R_{CONTRA}*) are the area under the curve of the trial-averaged ΔF/F response to the binocular (*BOTH*) and contralateral (*CONTRA*) trials, respectively. For the full-field flash stimulus the response modulation index was always calculated during the "On" period, for the buttons belonging to both On and Off populations, since the pupillary-mediated binocular modulation always occurred during the onset of the stimulus. For calculating the binocular modulation during the full-field contrast ramp stimulus, we used the area under the curve (AUC) of the cycles 2–4 to compute the RMI to the "low contrast phase" of the stimulus, while the AUC of the cycles 7–11 was used to calculate the RMI to the "high contrast phase" of the stimulus.

## Identification of functional populations

In order to separate the boutons according to the dynamics of their responses, we used dimensionality reduction to try to construct a low-dimensional embedding where boutons with similar time course are close together. To do so, we first selected for each bouton a "response signature". For recordings where the responses to both full-field flash and full-field contrast ramp were available, we concatenated the responses to the *CONTRA* trials of the two stimuli, and used it as the response signature. For the chirp stimulus, since it was specifically designed for maximally separating the RGC classes, we simply used the responses to the *CONTRA* trial. This created a response signature matrix NxT, where N is the number of boutons and T is the number of time samples of the response signature. The response signature matrix was then smoothed and each row independently normalized using z score normalization:

$$R_{norm} = \left[ \frac{R^T - \text{mean}(R)_t}{\text{std}(R)_t} \right]^T \quad (3)$$

where subscript *t* denotes the time dimension of the matrix *R* along which the operation has been calculated. Next, we applied PCA to the normalized matrix $R_{norm}$ to reduce the time dimension and obtained a new matrix NxT$_{reduced}$. For clustering analysis, we only kept the first two components with highest explained variance, and applied either k-means or Gaussian mixture models algorithm to the data in the 2-dimensional PCA space. The number of clusters was empirically tuned, and the purity of each cluster individually inspected. In the cases where PCA underperformed and failed in reaching a clear discrimination of the response dynamics, we applied the UMAP to reduce the time dimension. The clustering was then performed on the data projected onto the first 2 components found by UMAP, analogously to the clustering performed in the PCA space.

## SNR calculation

To compute the SNR, we estimated the signal and the noise components as previously described in ref. 41. We estimated the signal variance $V_{signal}$ of each bouton by averaging over the covariances calculated between all the $\binom{n}{2}$ combinations of trial repetitions:

$$V_{signal}(s) = \frac{1}{\binom{n}{2}} \sum_{i=1}^{n-1} \sum_{j=i+1}^{n} \text{Cov}(R_i, R_j) \quad (4)$$

where *n* is the number of trials for the stimulus *S*, and $R_i$ and $R_j$ are the responses to the *i*th and *j*th trial, respectively. The noise variance $V_{noise}$

was calculated as the difference between $V_S$ and the average across the within-trial variances calculated for all the repetitions. The SNR was then calculated by taking the ratio between $V_{signal}$ and $V_{noise}$.

$$V_{noise}(s) = V_{signal}(s) - \frac{1}{n} \sum_{i=0}^{n} \text{Var}(R_i) \quad (5)$$

$$SNR(s) = \frac{V_{signal}(s)}{V_{noise}(s)} \quad (6)$$

For estimation of the SNR for each peak of the contrast ramp separately, a subset of the full data was use (20% of top-responding boutons). This led to an increased noise in SNR calculations when calculated using the abovementioned method. In this case, we instead estimated SNR improvement by dividing mean of response to each contrast ramp step by standard deviation of the noise.

$$SNR = \frac{\mu_{signal}}{\sigma_{noise}} \quad (7)$$

## Statistical analysis and quantification

All the statistical comparisons between paired data were performed using paired *t-test*. Unless otherwise noted, all population data are reported as mean ± SEM, n represents the number of cells analyzed, and N represents the number of animals. Bonferroni correction was applied when multiple hypotheses were tested throughout. The distance between data distributions was quantified using the Kolmogorov–Smirnov test. Significance intervals for RMI distributions was estimated by taking the 95th percentile of the distribution fitted over the shuffled data. All tests and quantifications were performed using Matlab or python.

## Human psychophysics

**PLR kinetics.** *Subjects.* Three male human observers aged 25–35 participated in the study. One observer was one of the authors; the rest were naïve to the purposes of the study and received a small monetary compensation. All subjects had normal uncorrected vision. The study was conducted in accordance with the principles of the Declaration of Helsinki and the guidelines of the University of Helsinki ethical review board. The participants signed a written informed consent.

*Stimuli.* To approximately match the intensity of the light stimuli utilized in the mouse experiments in terms of photon captured by rod photoreceptors (R*) across mice and humans, the irradiance spectrum (in Watts m$^{-2}$ nm$^{-1}$) of the OLED screen utilized for murine experiments was calculated based on the apparent brilliance of the screen measured with a hand-held lux meter, and on the emission spectrum of the OLED screen weighted by the photopic luminosity function (in lumens W$^{-1}$, CIE 1978) [42]. Based on the irradiance spectrum of the screen, the number of R* per mouse rod per second was then estimated as described in ref. 43, considering the optic losses of intraocular media and the collecting area of mouse rods as a function of wavelength, peaking at 0.93 R* μm$^{-2}$ rod$^{-1}$ hv$^{-1}$ [44]. Based on these calculations, the intensity of a 470 nm light source producing full-field illumination by means of a Ganzfeld bowl was then adjusted to produce a similar number of R* per human rod, calibrated as described in ref. 45. The mean light intensity utilized for human experiments was 1500 R* rod$^{-1}$ s$^{-1}$. This light intensity was modulated to produce 3 s steps of light matching the relative intensity of those utilized in the murine experiments (1895 R* rod$^{-1}$ s$^{-1}$ and 3160 R* rod$^{-1}$ s$^{-1}$) or modulated by a square or sinusoid frequency which linearly increased to 100% contrast in 12 cycles.

*Apparatus*. The experiment was run in a pitch-black light-proof room utilizing a Ganzfeld bowl built in house coated with BaSo4-based paint. The light source consisted of a high-power LED with an emission spectrum peaking at 470 nm (Thorlabs, M415L4), further narrowed with a 10 nm full width half-maximum bandpass filter centered to 470 nm (Thorlabs, FBH470-10). The current passing through the LED was precisely controlled by an in house build high-dynamic range LED driver with temporal precision of 10 µs, controlled by a single-board computer (Raspberry Pi 4, Raspberry Pi Foundation) running custom Python and BASH software. The Pupillary Light Responses were recorded with an infrared camera (RPI HQ cam, Raspberry Pi Foundation) and an 850 nm infrared LED estimated to produce less than $10^{-4}$ R* rod$^{-1}$ s$^{-1}$. Digitized videos of the participants' face were analyzed with a custom Python algorithm to segment out the pupils, and then to determine the pupil radius.

*Procedure*. All subjects were dark adapted during 15 min in a pitch-black room at the beginning of every experiment, after which they were allowed to adapt to the 1500 R* rod$^{-1}$ s$^{-1}$ background for 5 min. Each stimulus was presented from dimmest to brightest or from slowest to fastest, and each stimulus was presented for 4–7 repetitions. The order in which the stimulus types (steps or frequency ramps) were presented varied across experiments.

**Visual detection experiment.** *Subjects*. 8 human subjects (4 male and 4 female) aged 20–40 participated in the study, all of them naïve to the purpose of the study. All subjects had normal or corrected vision. Informed consent was obtained by all participants as per EMBL BIAC (Bioethics Internal Advisory Committee), DPO (Digital Protection Office), and IP68 (EMBL Internal Policy, section 68) guidelines (approved study reference DPR-2023-008), and no financial compensation was given. Participant gender was not considered in this study, since the fundamental aspect of vision probed was not expected to have substantial sex differences. Researchers analyzing the data were blinded to subject identity, but not to experimental conditions (i.e., low versus high contrast condition) because the shape of the response would immediately break the blind during analysis. Volunteers were all lab members of EMBL Rome, and thus could confound results due to shared environment and educational background.

*Stimuli*. The stimulation protocol included a randomized sequence of 8, 3 s low-amplitude luminance step (9.5–12 LUX) and 8, 3 s high-amplitude luminance steps (9.5–65 LUX), matching the same luminance values used for the experiment we did in mice. Each trial consisted of 6 s of baseline luminance (9.5 LUX), followed by a 3 s-long, step-like luminance increase (to either 12 or 65 LUX). Importantly, the luminance during the 3 s step was kept constant until the end of the trial. Prior to the stimulation protocol start, the subjects were habituated to the baseline luminance (9.5 LUX) for 10 s.

*Task*. The subjects were informed they would have to perform a visual detection task. The task would consist of detecting minor screen luminance fluctuation happening randomly, at any point, during any of the luminance steps, regardless of the trial type (low or high amplitude luminance change). Subjects were asked to promptly report whenever they perceived the luminance fluctuation by pressing a keyboard key, and, importantly, they were not aware that all the luminance steps were made uniform in time. The subjects were allowed to practice in a trial round before performing the actual experiment, in order to get familiar with the task.

*Apparatus*. The visual stimuli were designed using PsychoPy[51] and presented on a 34.1″ LCD monitor (DELL U3419W). The subjects placed their head on a stabilizer device at 30 cm distance from the monitor. Pupil videography was performed using a NIR-optimized camera (Basler acA1300-60g) mounting a zoom lens (Navitar f = 100/F2.8) and the eye was illuminated using an IR lamp. Videos were acquired at 30 hz and the pupil was segmented using MEYE[52] and analyzed with a custom Python code.

## Reporting summary

Further information on research design is available in the Nature Portfolio Reporting Summary linked to this article.

## Data availability

The raw data that support the mouse findings of this study are available upon request, since the two-photon recordings are several terabytes large, and thus not compatible with any existing repository. The processed data of analyzed bouton activity have been deposited in github under accession code https://zenodo.org/records/15709769 (DOI: 10.5281/zenodo.15709769). No additional aspect of the human psychophysics data can be shared due to respective privacy policies. Any additional requests for material and correspondence should be addressed to SR. Source data are provided with this paper.

## Code availability

All software used and data after bouton dF/F extraction can be found in Github (https://github.com/tjasalapanja/PLR_RGC_analysis), or https://zenodo.org/records/15709769 (DOI: 10.5281/zenodo.15709769).

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

## Acknowledgements

We would like to thank all members of the Rompani, Asari, and Gross labs for their assistance with all experiments and helpful discussions. Facilities: Olga Boruc at the Laboratory Animal Resources (LAR), Neil Humphreys, Jim Sawitzke, and Adriana Caballero at EMBL Rome Gene Editing and Virus Facility (GEVF), Alvaro Crevenna at the Microscopy Facility, and Emerald Perlas at the Histology Facility. We thank Drs. Jayaraman, Kerr, Kim, Looger, and Svoboda and the GENIE Project, Janelia Farm Research Campus, HHMI, for GCaMP6 and GCaMP8, and the lab of Botond Roska for the ProA5 AAV promoter. Human face used in Figs. 7 and 8: NIAID Visual & Medical Arts. (10/7/2024). Female Upper Profile Straight On View. NIAID NIH BIOART Source. bioart.niaid.nih.gov/bioart/548. Funding for this project is primarily EMBL (internal EMBL funding 50500, SR), ETPOD (internal EMBL funding 50653, GF), FWO senior postdoc function (FWO grant number 1298724N, GF).

## Author contributions

Conceptualization: T.L., P.M., G.F., A.A. and S.R. Methodology: T.L., P.M., G.F., M.T., T.B., A.A., H.A., and S.R. Software: T.L., P.M., M.T., O.R., G.F., T.B., A.A., and H.A. Investigation: T.L., P.M., M.T., O.R., G.F., A.M., T.B., A.A., C.R., A.G., J.J., M.S., P.A., and G.A. Resources: J.J. Writing—Original draft: T.L. and S.R. Writing—review and editing: T.L., P.M., G.F., H.A. and S.R. Visualization: T.L., P.M. Supervision: G.F., G.K., H.A., P.A., and S.R. Project administration: S.R. Funding acquisition: G.K., H.A. and S.R.

## Funding

## Competing interests

The authors declare no competing interests.
