## [Transparent Peer Review file · Nature Communications]

Pupil size modulation drives retinal activity in mice and shapes human perception

Corresponding Author: Dr Santiago Rompani

Version 0:

Reviewer comments:

Reviewer #1

(Remarks to the Author)

Visual perception relies on the detection of light and the subsequent processing of information by the retina and brain. Modulation of the visual input starts before the retina, at the pupil. The pupil can change its size based on the luminance of the environment, contrast, state of arousal, or mental effort, thereby affecting the amount of light reaching the retina. However, the ability for pupillary change to induce retinal activity has not been tested. Tjasa Lapanja and his colleagues in the manuscript titled "Pupil-induced retinal binocularity potentiates low contrast vision", found that pupillary constriction due to increased light, the pupillary light reflex (PLR), can drive strong responses in the retinal output neurons, the retinal ganglion cells (RGCs) in vivo in mice. Furthermore, the consensual PLR allows one eye to indirectly respond to luminance changes happening on the other eye, leading to a binocular response and binocular modulation consistent with the potentiation of low contrast vision. These findings seem to have a certain degree of novelty. However, as mentioned by the authors in the manuscript, these are caused by changes in light luminance entering the retina due to PLR, making the observed phenomena in this manuscript quite obvious. In other words, the constriction of the pupil can cause changes in the amount of light entering the eye, thereby inducing the response of OFF RGCs. Although the authors believe that PLR-driven RGC activities support binocular facilitation, these are predicted. If this work is to be worthy of publication in Nature Communications, the authors need to address how these responses affect the visual perception process, namely the function at the level of cortical response processing, animal visual behavior, and human visual psychophysical level.

1. The authors describe that during light-induced pupil constriction, RGC boutons in the dLGN exhibit a non-canonical PLR-driven response. They refer to the non-canonical response component of OFF RGCs during the flash as a delayed-ON response. This response seems to be an OFF response of the OFF RGCs themselves, caused by the reduction in light luminance due to pupil constriction during continuous illumination. Similarly, as shown in Figure S1A-S1B, the boutons of ON RGCs in the dLGN also exhibit a baseline decrease in background light-induced ON response due to reduced light luminance during ipsilateral eye stimulation. Based on the existing data, we are not clear about how the increase in OFF responses and decrease in ON responses induced by pupil constriction are processed in the visual cortex and what impact they have on visual function.
2. In Figure 4, the authors characterized that pupil-induced RGC activity is relayed to the visual cortex. However, in Figure 4D, we cannot distinguish the PLR-induced responses elicited by ipsilateral eye stimulation from the existing results. This is because the dLGN neurons projecting to V1 receive input from RGCs of both eyes, and light stimulation of the ipsilateral eye may also elicit ON responses in dLGN neurons. The authors should consider how to differentiate between these two types of responses.
3. The authors demonstrate in Figures S6A and S6C that there are two populations of contralateral and bilateral responses induced by the pupil. However, in Figures 5B and 5D, it appears that only the population from Figure S6C was selected for analysis and presentation. The authors need to clearly explain the reason for this choice. This could lead to confusion for the readers, as Figures 5B and 5D suggest that under high contrast light stimulation conditions, no pupil-driven ON response was induced. Additionally, it can be observed from Figures S6A and S6C that there are no changes in the classic OFF response peak of OFF RGCs with different light intensity stimuli. However, Figures 6C and 6D show that an increase in light intensity causes changes in the response amplitude.
4. In addition to providing the illuminance of light intensity for the low contrast and high contrast sections, the authors should

also clearly present the illuminance values of light intensity in the figures for other parts of the experiments. This would allow readers to assess the conditions of each light stimulus.

5. Meanwhile, the authors should analyze the magnitude of the pupil-driven delayed-ON response under different light intensities to determine how these responses are affected by pupil constriction under various light stimuli.

6. In Figure 2, the authors should test the pupil-driven delayed-ON response elicited by direct light stimulation without the screen background gray light condition. This would more clearly demonstrate and explain that the response in the contralateral eye during ipsilateral eye stimulation is caused by the reduction of background light intensity entering the retina.

7. Throughout the manuscript, various conditions of stimulation induce different responses in varying proportions of RGC boutons. In addition to presenting the pupil-driven delayed-ON response, responses that do not exhibit this characteristic should also be included in supplementary figures. These figures should illustrate the proportional relationships.

8. The authors should proofread the manuscript for any language errors. For example, there is no Fig. S1J, and in the supplementary material line 367, "start" is not capitalized. In the main text, line 162, "(cite)" does not have any citation content.

Reviewer #2

(Remarks to the Author)

Reviewer #3

(Remarks to the Author)

Lapanja et al. investigated whether the pupillary light reflex (PLR) modulates retinal activity as a non-canonical form of binocularity. The study uses leading-edge imaging techniques (calcium signals of labeled RGC axonal boutons in dLGN) to propose a new role for pupil dynamics and their effects on non-canonical binocular responses in (mainly OFF) RGCs. The study shows that this PLR-mediated activity is relayed to the visual cortex, that it is most pronounced with low contrast stimulation, that it is potentiated with binocular stimulation, and that the pupil dynamics of mice match those of humans. The authors conclude that the PLR-mediated activity likely plays a role in shaping visual processing. Overall, there are exciting findings here with intriguing implications for an understudied topic (namely, implications for visual processing due to the PLR itself). Additionally, the results have implications for the role of the pupil in cortical visual processing and, potentially, visual disorders. However, some important experimental controls, clarifications, and analyses should be included to strengthen the conclusions and implications of the paper.

1. The study shows that the consensual PLR itself induces retinal responses. While this conclusion is supported by the pharmacological experiments on PLR, there remain several important questions about the underlying mechanism that should be addressed experimentally, and that would add significant strength to the conclusion. Consider the following:

a. Is background luminance necessary for the PLR effect? If the ipsilateral eye was stimulated while the contralateral eye was exposed to a black screen, would the PLR response still be induced? Would this response be different at the V1 level where there may be more binocular integration? It is stated that background light is crucial for the consensual PLR response (Lines 157-158) but this was not shown. This seems important to show rigorously (e.g., showing how the effect varies as a function of background luminance).

b. Could the stimulus itself increase arousal or activate neuromodulation of the RGC axons (e.g., Liang et al, 2020; Reggiani et al, 2023)? This is a critical component to rule out in this context, particularly since OFF RGCs seem most sensitive to this. This concern is perhaps also related to the fact that there are significant differences in the strength of the effects in anesthetized versus awake mice (l.113). For example (one experiment, but not the only one) could be to block effects of neuromodulator release potentially related to flash-induced arousal changes, and showing the PLR mediated effects persist. This would greatly strengthen the interpretation that this is purely an effect due to aperture size and photons.

c. If the "delayed-ON response" is due to a reduction in the number of photons on the retina, then can this same response be induced by presenting a "dimming" stimulus that mimics what occurs during a pupil constriction? Would this response then occur for ON and OFF axons, even in the presence of atropine? Such evidence would complement the current findings and help sharpen evidence for the mechanism driving the response (pupillary mediated decrease in photons).

2. The term "delayed-ON PLR response" seems to be misrepresenting this signal, risking confusion for readers. While the response does occur during light onset, it is still a response to a reduction of retinal photons (due to pupil constriction) and therefore more accurately described as a "non-canonical" OFF signal. We suggest using this terminology throughout the paper, since this centers the description on the cells being measured (OFF cells, mainly), while also capturing the novel aspect of the signal.

3. Moreover, it is intriguing that this pupil constriction (reduction in retinal photons) would lead to a similar increase in neural activity in ON RGCs (Figure 3). What is the explanation for this? It seems there may be mixing of ON and OFF channel information with pupil changes, or perhaps this is related to mixed selectivity of the imaged axons? Or a dependence on background luminance regimes? A more in-depth analysis and discussion of these mixed ON and OFF responses due to pupil constriction, and their implications for cortical ON and OFF processing, is needed.

4. There are some major differences between ipsi and contra stimulation that should be highlighted and reported throughout:
 - a. In Figure 2b and d, the delayed-ON response is larger and slower for ipsi than contra, but the pupil constriction is significantly smaller when the ipsi eye was stimulated. How can this be explained since the greater pupil constriction of the contra eye did not induce an equivalent response?
 - b. Was the delayed-ON response time course significantly different between ipsi and contra? The time histograms are plotted (for example Figure 2c) but the data is not mentioned in the results.
 - c. Are the pupil dynamics different between ipsi vs contra stimulation? In Figure 2g it appears that amplitude is different but what about timecourse?
 - d. Why is the ipsi-contra relationship opposite in Figure 6 when a sinusoidal stimulus is used (Figure 6c and d – contra greater than ipsi). Why do the authors think this is?
5. The latency of the delayed-ON response appears to be different in ON and OFF RGCs (Figure 3). Is this consistent? The timing differences of the responses might also be worth reporting more clearly. Moreover, are delayed-ON responses from binocular stimulation faster than monocular stimulation in both ON and OFF cells (Figure 5)?
6. In Figure 3, ON RGCs clearly show an ipsi delayed-ON response to luminance step at the start of the chirp stimulus. However, in Figure S1, the ipsi stimulus does not appear to drive a delayed-ON response despite the text stating so (lines 114-115). Please clarify.
7. In Figure 5, contra and both eyes are compared. But ipsi induced responses appear to be larger with the flash stimulus (Figure 2). Is the same binocular potentiation seen when comparing ipsi and both eyes?
8. Were ON RGCs analyzed in the Figure 2 and Figure 5 experiments? If so, do they show similar effects? Moreover, are the cells in the Figure 6 and 7 experiments ON or OFF (or combined)? Is the same binocular facilitation seen in ON RGCs?
9. The Figure S6 caption states that only the subpopulation seen in Figure S6c was used for further analysis throughout the study. However, the subpopulations in Figure S6a also shows potentiation. Are the same results seen with these other subpopulations?
10. In Figure 6, one temporal frequency that the pupil could track was tested (0.55 Hz). Would other slow temporal frequencies produce the same results? Were any others tested?
11. In Figure 8, which eye were pupil responses in human subjects measured and which eye was stimulated? The important comparison here would be stimulation of one eye and measurement of the pupil response in the other eye (the main effect of the study), which seems important to add here (or discuss, at the very least). Is the optimal temporal frequency for pupil stimulation the same between mice and humans?
11. ON pathways (photoreceptor-mediated and melanopsin) control the pupil response. Would be a similar inhibition of the PLR response occur with APB application (blocking ON pathways)? If the delayed-ON response signals a global state change in luminance, is melanopsin involved?
12. Many details concerning the anesthesia/awake status, numbers of mice or recordings per result, and information regarding which boutons were analyzed are missing and should be included throughout the results and in the figure captions. Just to state one example, in the paragraph starting at lines 168 and 195, and in the Figure 4 caption. Please double check this information throughout the paper and report the relevant details so readers can fully interpret the experimental conditions, sample sizes, etc. for each result.
13. Further, it would be helpful to provide more details about the bouton analysis – are these from different RGCs, or the same? Can this be disambiguated (e.g., using trial-by-trial variability / correlations)? It seems important to clarify and potentially correct if this poses statistical repeated measures (i.e., all 8 OFF boutons in a FOV are from the same RGC, versus 8 different RGCs).
14. Related to the above, it would be highly beneficial to report how statistically reliable the effects are within subject. For example, is the strength / frequency of “delayed ON responses” similar across mice? Across experiments within a mouse? Such details would enhance the interpretability of findings and illustrate their generality (or not).

Minor:

15. Figure 1. In panels E and F, the simultaneous pupil traces should be included. Are these examples just one bouton or an average of many boutons?
16. Figure 2g. Please plot the ipsi and contra pupil traces on the same timescale – the ipsi response is much smaller. Consider also for Figure 6c and d.
17. What flash intensity was used in Figures 1 and 2?
18. Figures 3 and 4. Please include the pupil traces for panels C and D.
19. What was the ambient brightness of the experimental room (i.e. were both rod and cone pathways active)? Depending on the RGCs labeled, adaptation state may influence the measured retinal activity.
20. The last paragraph in the discussion suggests this could be related to a global state-change signal to the brain. How do the authors think this increase in activity in OFF RGCs during a light flash would lead to this effect? Are there other naturalistic stimulus conditions that could induce these delayed-ON responses and binocular facilitation?
21. In Fig. S4, there appears to be a mismatch in the timing of the chirp stimulus schematic at the top and the pupil traces

(e.g. the second pupil constriction event begins before the luminance increase). Please check or explain.

22. Line 162. It appears a citation is missing.

23. Lines 316-318. While the SNR values may be consistent, the response latencies of the effects here and binocular summation for reaction times are drastically different (Blake, Martens, Gianfilippo, IOVS 1980). This should be mentioned, or the line of argumentation dropped.

24. Line 332. What is the definition of the “dominant eye” in this context?

25. In Figure 7 it would be helpful to label the graphs with what they are showing (e.g. top 20%, etc).

26. L.367 – It seems overly speculative to claim that a conserved delay in PLR kinetics (mainly due to stereotyped neuromuscular kinetics) as evidence for a “conserved mechanism” of error correction of luminance changes (which is not explicitly shown in this study). Please revise or clarify these speculations, or simply remove them as they seem misplaced given the findings of the study.

27. Line 277 and Figure 7. What if all boutons were analyzed?

28. Figure 6 title. “constraint” should be “constrain”.

Reviewer #4

(Remarks to the Author)

Version 1:

Reviewer comments:

Reviewer #1

(Remarks to the Author)

I appreciate the authors' additional experiments, revisions to the manuscript, and detailed responses to the reviewers. Although the visual response induced by PLR does not show significant physiological meaning in human psychophysical experiments, this article currently presents a complete and systematic scientific story.

Reviewer #2

(Remarks to the Author)

Reviewer #3

(Remarks to the Author)

The authors have done a significant amount of work to address my concerns, along with addressing related concerns from other Reviewers. The overall conclusions are strengthened by the additional experiments about background luminance dependence (S3), and simulated dimming with dilated pupil (S8). The line of argumentation focusing on low contrast SNR improvements with binocular facilitation is also stronger (S17). The addition of animal-by-animal variability data are helpful for readers to judge the effects for themselves. Overall, the authors are commended for their improved rigor and stronger support for the conclusions of this very interesting study highlighting novel aspects of binocular PLR interactions.

Reviewer #4

(Remarks to the Author)

Reviewer #1 (Remarks to the Author):

Visual perception relies on the detection of light and the subsequent processing of information by the retina and brain. Modulation of the visual input starts before the retina, at the pupil. The pupil can change its size based on the luminance of the environment, contrast, state of arousal, or mental effort, thereby affecting the amount of light reaching the retina. However, the ability for pupillary change to induce retinal activity has not been tested. Tjasa Lapanja and his colleagues in the manuscript titled “Pupil-induced retinal binocularity potentiates low contrast vision”, found that pupillary constriction due to increased light, the pupillary light reflex (PLR), can drive strong responses in the retinal output neurons, the retinal ganglion cells (RGCs) in vivo in mice. Furthermore, the consensual PLR allows one eye to indirectly respond to luminance changes happening on the other eye, leading to a binocular response and binocular modulation consistent with the potentiation of low contrast vision. These findings seem to have a certain degree of novelty. However, as mentioned by the authors in the manuscript, these are caused by changes in light luminance entering the retina due to PLR, making the observed phenomena in this manuscript quite obvious. In other words, the constriction of the pupil can cause changes in the amount of light entering the eye, thereby inducing the response of OFF RGCs. Although the authors believe that PLR-driven RGC activities support binocular facilitation, these are predicted. If this work is to be worthy of publication in Nature Communications, the authors need to address how these responses affect the visual perception process, namely the function at the level of cortical response processing, animal visual behavior, and human visual psychophysical level.

We addressed the point mentioned at the end of this paragraph in item 1 just below, namely by having performed new experiments addressing cortical response processing and human psychophysics.

1. The authors describe that during light-induced pupil constriction, RGC boutons in the dLGN exhibit a non-canonical PLR-driven response. They refer to the non-canonical response component of OFF RGCs during the flash as a delayed-ON response. This response seems to be an OFF response of the OFF RGCs themselves, caused by the reduction in light luminance due to pupil constriction during continuous illumination. Similarly, as shown in Figure S1A-S1B, the boutons of ON RGCs in the dLGN also exhibit a baseline decrease in background light-induced ON response due to reduced light luminance during ipsilateral eye stimulation. Based on the existing data, we are not clear about how the increase in OFF responses and decrease in ON responses induced by pupil constriction are processed in the visual cortex and what impact they have on visual function.

To address the reviewer’s concern about the role of PLR-driven visual responses in the visual cortex and visual function, we performed two additional experiments: 1) we recorded PLR-driven visual responses in mouse V1 neurons, and 2) we performed human psychophysics to

probe if the PLR can induce a perceptual effect consistent with the properties of PLR-induced responses we found in mice.

PLR-driven visual responses in mouse V1 (new Fig. 4F-N): In order to record PLR-driven activity in the visual cortex, we performed *in vivo* 2-photon calcium imaging in layer 2/3 of V1 in awake animals. We recorded the activity of neurons in the monocular zone in one hemisphere, while a grey screen was placed in front of the contralateral eye. To induce consensual PLR, we stimulated the ipsilateral eye with an LED optic fiber. In two animals, we observed 150 cells that showed either strong positive or negative correlation to the induced pupil constriction in at least 1 trial, suggesting that V1 neurons can also respond to pupillary constriction. This is unlikely due to canonical binocular responses in cortex since we recorded from the monocular region of V1 and the ipsilateral responses were 1-second delayed, consistent with PLR kinetics.

These findings are additionally addressed in the results and discussion, and overall, further bolster the claim that these PLR-induced responses are substantial and perceived.

Human psychophysics: We hypothesized that PLR would transiently reduce the perceived brightness of a uniform stimulus (a grey screen flashing white for 5 seconds, then returning to grey) due to the activation of the OFF channel during the PLR event, an effect we called "perceptual dimming". Additionally, based on data from mouse RGC responses, we predicted the perceptual dimming to be more prominent during low-amplitude luminance changes than during high-amplitude luminance changes, despite the pupil constricting more in high contrast than low contrast. To test this hypothesis, we presented human volunteers with uniform low- or high-contrast full-field flash stimuli and asked them to report any change in the perceived brightness after the onset of the stimulus, finding that volunteers indeed exhibited the predicted perceptual dimming (Fig. 8). We think this perceptual dimming is due to PLR-induced responses for three reasons: 1) Its kinetics match the one of the PLR, taking about 1 second to occur (Fig. 8F). 2) It was more salient in low contrast than high contrast, as predicted by our mouse data (Fig. 8B-E, compare to Fig. 5), and 3) Higher detection of perceptual dimming in low contrast was seen despite less absolute pupillary constriction than at high contrast, as was observed in mice (Fig. 8, compare to for instance Fig. 5D). These three similarities to how the PLR-induced responses behave in the mouse strongly suggest perceptual dimming in humans is also due to PLR-induced responses.

Overall, these two new experiments suggest that the PLR-induced response not only is present in the visual cortex, but is also actively perceived in humans, which we believe increase this study's impact and thus suitability for Nature Communications, as the reviewer mentioned in their summary paragraph.

Further discussion on this psychophysics was added to the main text, and previous Figure 7 was moved to the supplemental figures (Fig. S17) in order to make room for new Figure 8. A minor refinement on the LGN axons projecting to V1 was also added to new Fig. 4A-D, to better show how those responses mirror the PLR-induced response, with previous Figure 4 now in the supplement (Fig. S11).

2. In Figure 4, the authors characterized that pupil-induced RGC activity is relayed to the visual cortex. However, in Figure 4D, we cannot distinguish the PLR-induced responses elicited by ipsilateral eye stimulation from the existing results. This is because the dLGN neurons projecting to V1 receive input from RGCs of both eyes, and light stimulation of the ipsilateral eye may also elicit ON responses in dLGN neurons. The authors should consider how to differentiate between these two types of responses.

The reviewer makes an important point regarding our claim on PLR-driven responses in the LGN. Our data cannot exclude the presence of binocular LGN neurons. We changed the main text to address this point more clearly. However, while LGN binocularity cannot be entirely excluded, our results could not be explained by canonical LGN binocularity and are consistent with PLR-driven LGN responses for four reasons:

1) *Bauer et al.* (2021), reference 36, studied the binocularity of the LGN neurons and concluded that the LGN boutons in V1 exhibit barely any functional binocularity (mean ODI index 0.95, with 1 = a fully monocular response). They concluded 7% of boutons exhibit binocular responses to full-field stimulation. With lowering of threshold this value could be increased to 21%. However, they would have missed any PLR-driven binocularity, as they were covering the opposite eye, an equivalent experiment to what we report in Fig. S3, where lack of background illumination removes PLR-induced responses (an experiment requested in later point 6 as well). In our LGN axon recordings, we used stringent criteria of $RQI > 0.3$ to determine contralateral LGN boutons responsive to ipsilateral stimulation. When presenting full-field flash, 56.1% of boutons responded binocularly, more than twice the least stringent estimate from *Bauer et al.* and 8 times more than their stringent estimate, which matches the criteria we used as our threshold for claiming binocularity.

2) Responses to ipsilateral stimulation of OFF LGN boutons occur with temporal characteristics in line with responses to PLR (new Fig. 4E, bottom). Namely, the response peaks at 0.9s after the onset of the stimulus, which is after the start of PLR. In comparison, the canonical OFF response peaks at 0.56 s after the offset of the stimulation, with many responses happening earlier (new Fig. 4E, top). Importantly, we did not observe any responses with characteristics of canonical ipsilateral responses (either ON or OFF responses, without the prolonged delay) that passed our RQI threshold. Furthermore, during ipsilateral chirp stimulation, responses were present during full-field parts of the stimulation but not during a high-frequency ramp, when the pupil fails to constrict (previous Fig. 4, now Fig. S11).

3) All the LGN boutons clusters reported in new Fig. 4A-E showed “pure” OFF or ON responses when stimulated contralaterally (new Fig. 4C, left). If the drastic qualitative change in the response property (e.g the canonical OFF response getting a non-canonical OFF response from a true ON cell) was due to RGCs axons convergence, that would imply that the same LGN neuron is contacted by OFF-only contralateral RGCs and ON-only ipsilateral RGCs, a functional convergence that was not observed by two other studies specifically looking for such drastic cell-type convergence in the LGN (references 32 and 33).

4) Our additional experiments in V1 (new Fig 4F-N) are consistent with the PLR-driven responses being relayed from the LGN to cortex. The human psychophysics experiments (new Fig. 8) further support that PLR-driven responses reach V1.

3. The authors demonstrate in Figures S6A and S6C that there are two populations of contralateral and bilateral responses induced by the pupil. However, in Figures 5B and 5D, it appears that only the population from Figure S6C was selected for analysis and presentation. The authors need to clearly explain the reason for this choice. This could lead to confusion for the readers, as Figures 5B and 5D suggest that under high contrast light stimulation conditions, no pupil-driven ON response was induced. Additionally, it can be observed from Figures S6A and S6C that there are no changes in the classic OFF response peak of OFF RGCs with different light intensity stimuli. However, Figures 6C and 6D show that an increase in light intensity causes changes in the response amplitude.

In order to address the reviewer's concerns, We included new Fig. S13 for OFF cells and new Fig. S14 for ON cells, by including all the quantification of the binocular facilitation presented in figure 5, computed on all the boutons, divided in ON and OFF macro populations. For the main text, we decided to keep the quantification only of the first OFF population as we think it's the most effective in explaining to the reader the phenomena that we are describing, while the effect of this phenomena is still evident when analysing all the boutons together (Fig. S13C). Concerning the point of the OFF response peaks not scaling with the intensity of the 5 light level stimulus (see also Fig S12) compared to the contrast ramp, the conditions are not quite comparable: the baseline in the 5 light levels is much darker compared to the amplitude ramp, which oscillates from a median grey value, and the amplitude ramp starts at a lower change in contrast compared to the first light level in the five flashes.

4. In addition to providing the illuminance of light intensity for the low contrast and high contrast sections, the authors should also clearly present the illuminance values of light intensity in the figures for other parts of the experiments. This would allow readers to assess the conditions of each light stimulus.

The light levels have been added to the figures where high and low contrast conditions are compared to each other (Fig. 5, Fig. S13-S14). For other figures, we added them to the figure legends.

5. Meanwhile, the authors should analyze the magnitude of the pupil-driven delayed-ON response under different light intensities to determine how these responses are affected by pupil constriction under various light stimuli.

To address the reviewer's suggestion, we included Fig. S13, showing maximal df/f (Fig. S13A) and activity heatmaps (Fig. S12) of the PLR-driven responses calculated for all the OFF boutons, in all the 5 light intensities conditions, for contra, ipsi and binocular presentation of the stimuli.

6. In Figure 2, the authors should test the pupil-driven delayed-ON response elicited by direct light stimulation without the screen background gray light condition. This would more clearly demonstrate and explain that the response in the contralateral eye during ipsilateral eye stimulation is caused by the reduction of background light intensity entering the retina.

In Fig. S3, we show the results of an experiment where we presented mice with ipsilateral, contralateral, and binocular stimulation, either in darkness or with background stimulation. We observe ipsilateral responses in both ON and OFF boutons, binocular facilitation in OFF boutons, and the change in the shape of ON responses in the presence of grey background stimulation. None of the responses was observed with a black background stimulation. To perform this experiment with additional accuracy, we decided not to use the 2-screens setup (for which complete darkness cannot be achieved), but a domed array of LEDs covering each eye, further ensuring that the condition lacking a background was indeed fully dark.

7. Throughout the manuscript, various conditions of stimulation induce different responses in varying proportions of RGC boutons. In addition to presenting the pupil-driven delayed-ON response, responses that do not exhibit this characteristic should also be included in supplementary figures. These figures should illustrate the proportional relationships.

To further present the variability of PLR-driven responses, we include the data from all RGC subtypes, including ON boutons, where the effect of PLR is often subtler (Fig S12). We also further explore PLR-driven responses during low- and high-contrast stimulation (Fig. S12-S13). Another important aspect of variability in the strength of PLR-driven responses is inter-animal variability. Anaesthesia can have a variable effect on pupil constriction (see ref. 30), impacting the strength of PLR and, consequently, the intensity of PLR-driven responses. In Fig. S4, we are presenting all OFF RGC responses in 5 animals and comparing them to the strength of the PLR in each recording. The data shows that the strength of constriction is an important factor determining the prevalence and strength of the PLR-driven retinal activity, and even when including the animals that had not optimal PLR, the effects we reported throughout the paper were still very statistically significant.

8. The authors should proofread the manuscript for any language errors. For example, there is no Fig. S1J, and in the supplementary material line 367, "start" is not capitalized. In the main text, line 162, "(cite)" does not have any citation content.

Mentioned errors were corrected and further proofreading was conducted.

Reviewer #3:

Lapanja et al. investigated whether the pupillary light reflex (PLR) modulates retinal activity as a non-canonical form of binocularity. The study uses leading-edge imaging techniques (calcium signals of labeled RGC axonal boutons in dLGN) to propose a new role for pupil dynamics and their effects on non-canonical binocular responses in

(mainly OFF) RGCs. The study shows that this PLR-mediated activity is relayed to the visual cortex, that it is most pronounced with low contrast stimulation, that it is potentiated with binocular stimulation, and that the pupil dynamics of mice match those of humans. The authors conclude that the PLR-mediated activity likely plays a role in shaping visual processing. Overall, there are exciting findings here with intriguing implications for an understudied topic (namely, implications for visual processing due to the PLR itself). Additionally, the results have implications for the role of the pupil in cortical visual processing and, potentially, visual disorders. However, some important experimental controls, clarifications, and analyses should be included to strengthen the conclusions and implications of the paper.

1. The study shows that the consensual PLR itself induces retinal responses. While this conclusion is supported by the pharmacological experiments on PLR, there remain several important questions about the underlying mechanism that should be addressed experimentally, and that would add significant strength to the conclusion. Consider the following:

a. Is background luminance necessary for the PLR effect? If the ipsilateral eye was stimulated while the contralateral eye was exposed to a black screen, would the PLR response still be induced? Would this response be different at the V1 level where there may be more binocular integration? It is stated that background light is crucial for the consensual PLR response (Lines 157-158) but this was not shown. This seems important to show rigorously (e.g., showing how the effect varies as a function of background luminance).

To address the first, second and last questions posed, as also mentioned in item 6 of reviewer 1, in new supplementary figure 3 we show the results of an experiment where we presented mice with ipsilateral, contralateral, and binocular stimulation, either in darkness or with background stimulation. We observe ipsilateral responses in both ON and OFF boutons, binocular facilitation in OFF boutons, and the change in the shape of ON responses in the presence of grey background stimulation. None of the responses were observed without the grey background stimulation, when the background luminance was black. To perform this experiment with additional accuracy, we decided not to use the 2-screens setup (for which complete darkness cannot be achieved), but a domed array of LEDs covering each eye, further ensuring that the condition lacking a grey background was indeed fully dark.

To address the third question regarding V1, as mentioned from item 1 of reviewer 1: In order to record PLR-driven activity in the visual cortex, we performed *in vivo* 2-photon calcium imaging in layer 2/3 of V1 in awake animals. We recorded the activity of neurons in the monocular zone in one hemisphere, while a grey screen was placed in front of the contralateral eye. To induce consensual PLR, we stimulated the ipsilateral eye with an LED optic fiber. In two animals, we observed 150 cells that showed either strong positive or negative correlation to the induced pupil constriction in at least 1 trial, suggesting that V1 neurons can also respond to pupillary constriction. This is unlikely due to canonical binocular responses in cortex since

we recorded from the monocular region of V1 and the ipsilateral responses were 1-second delayed, consistent with PLR kinetics.

b. Could the stimulus itself increase arousal or activate neuromodulation of the RGC axons (e.g., Liang et al, 2020; Reggiani et al, 2023)? This is a critical component to rule out in this context, particularly since OFF RGCs seem most sensitive to this. This concern is perhaps also related to the fact that there are significant differences in the strength of the effects in anesthetized versus awake mice (1.113). For example (one experiment, but not the only one) could be to block the effects of neuromodulator release potentially related to flash-induced arousal changes and showing the PLR-mediated effects persist. This would greatly strengthen the interpretation that this is purely an effect due to aperture size and photons.

We believe that neuromodulation cannot play a substantial role in the PLR-induced responses we observed for 3 reasons:

1) The majority of our recordings were performed in anesthetized animals, where arousal couldn't play an important role—due to the induced anaesthetic state. While the data collected in awake animals is noisier, we believe that is largely due to motion artefacts, not to the role of neuromodulation in the awake condition. Furthermore, the main characteristics of PLR-driven responses, namely the response delay (Fig. 2A-E) and the binocular facilitation (Fig. 6I-K), remained unchanged between awake and anesthetized recordings.

2) Our experiments involving atropine and carbachol to remove PLR-driven responses in anesthetized animals strongly suggests a causal role for the pupil itself rather than an effect of neuromodulation. Local application of these pharmacological agents is not likely to affect any central neuro-modulatory system, yet these compounds quite thoroughly abolished the PLR-driven responses and binocular facilitations.

3) In Fig. S8, we show that the change of luminance alone is enough to cause the observed responses. As the reviewer suggested in the next point, we modelled the change of luminance on the retina due to the PLR and presented the PLR-adjusted stimulus to an animal with a fully dilated pupil. We show that with modelling the change of luminance on the retina due to PLR, we can recreate both PLR-driven binocular responses and binocular facilitation, which further confirm the direct, causal role of pupil size in driving the responses we observed.

c. If the “delayed-ON response” is due to a reduction in the number of photons on the retina, then can this same response be induced by presenting a “dimming” stimulus that mimics what occurs during a pupil constriction? Would this response then occur for ON and OFF axons, even in the presence of atropine? Such evidence would complement the current findings and help sharpen evidence for the mechanism driving the response (pupillary mediated decrease in photons).

We conducted this "dimming" experiment, as suggested by the reviewer. Fig. S8 shows that the dimming stimulus very well recapitulates the characteristics of both ON and OFF PLR-driven responses, as well as binocular facilitation.

2. The term "delayed-ON PLR response" seems to be misrepresenting this signal, risking confusion for readers. While the response does occur during light onset, it is still a response to a reduction of retinal photons (due to pupil constriction) and therefore more accurately described as a "non-canonical" OFF signal. We suggest using this terminology throughout the paper, since this centers the description on the cells being measured (OFF cells, mainly), while also capturing the novel aspect of the signal.

We agree with the reviewer's point and we changed this terminology throughout the manuscript.

3. Moreover, it is intriguing that this pupil constriction (reduction in retinal photons) would lead to a similar increase in neural activity in ON RGCs (Figure 3). What is the explanation for this? It seems there may be mixing of ON and OFF channel information with pupil changes, or perhaps this is related to mixed selectivity of the imaged axons? Or a dependence on background luminance regimes? A more in-depth analysis and discussion of these mixed ON and OFF responses due to pupil constriction, and their implications for cortical ON and OFF processing, is needed.

Indeed without a zoomed-in view of the responses to ipsilateral chirp in figure 3D, it may appear that the ON response also has an increase in activity similar to the OFF cells. However, when looking at the dynamics of the response in more detail (added Fig. F insert), it is clear the ON cells have an initial decrease in activity coincident to when the OFF cells have an increase. However, unlike the OFF cells, the decreased activity during a constriction phase of the PLR is followed by an increase in the activity, due to the rebound dilation of the pupil happening in the late phase of the PLR. This is further clarified in new Figure 3G, where we plot the time at which the PLR-driven responses reach their maximum (for both ON and OFF cells) or their minimum (only for ON cells). This plot better highlights the different behaviour of OFF and ON cells, with the latter showing negative modulation during the constriction phase followed by a positive modulation during the rebound-dilation phase. These findings are largely consistent with how ON and OFF cells should behave in this situation, so we believe the alteration of the figure largely solve this issue.

4. There are some major differences between ipsi and contra stimulation that should be highlighted and reported throughout: a. In Figure 2b and d, the delayed-ON response is larger and slower for ipsi than contra, but the pupil constriction is significantly smaller when the ipsi eye was stimulated. How can this be explained since the greater pupil constriction of the contra eye did not induce an equivalent response?

The reviewer makes a good point, which connects the three main variables underlying the process we describe in the paper: 1) the amplitude of the luminance change, 2) the current sensitivity of the individual RGC (which also depends on the state of adaptation of the

photoreceptors) and 3) the amplitude of the negative modulation in the number of photons reaching the retina due to the pupillary constriction.

When the contralateral eye is directly stimulated, the photoreceptors experience an abrupt change in the ambient luminance. We hypothesise that while the amplitude of the PLR event is directly proportional to the amplitude of the luminance change, if the retinal responses saturate then the decrease in the number of photons due to a PLR event might still not be enough for generating a retinal response, as the overall luminance is still above the upper bound of the sensitivity range of the adapted photoreceptors. When stimulating ipsilaterally, even if the amplitude of the PLR might be lower than the one elicited by the contralateral stimulation (which we hypothesise could be an effect of anaesthesia), the photoreceptors and RGC responses are not saturated, and relatively smaller changes in the number of photons reaching the retina can be detected, as they fall within the current sensitivity range defined by the adaptation states. In other words, we propose that the PLR-driven RGC responses are not proportional only to the absolute change in the number of photons reaching the retina, rather it is a combination of the latter and the current adaptation state of the retina. This has been briefly mentioned when emphasizing that a smaller pupil change can cause greater responses, despite that seeming paradoxical.

b. Was the delayed-ON response time course significantly different between ipsi and contra? The time histograms are plotted (for example Figure 2c) but the data is not mentioned in the results.

Overall there is no significant difference between the onset of the response in contra and ipsi stimulation, as is also plotted in new Fig. S1C-F. The peak of the response is reached slightly slower during ipsilateral stimulation (Fig. S1F), perhaps due to the fact that the response is stronger. This new data presentation hopefully makes that comparison more immediately apparent.

The one thing that does affect response dynamics is the animal-to-animal variability affecting the dynamics of the PLR. In Fig. S4 we can see a comparison between PLR in different animals as well as the responses of their RGC boutons. In animal 3 (green) for example, the PLR was slower than in other animals. We hypothesise that it could be an effect of anaesthesia. In this case, PLR driven responses (both to ipsi and binocular stimulation) in that animal were also delayed. However, despite such variability in our dataset, the statistical significance of our findings was not impacted.

c. Are the pupil dynamics different between ipsi vs contra stimulation? In Figure 2g it appears that amplitude is different but what about timecourse?

As shown in Fig S4E-F, there is no significant difference between the time-course of the PLR. However, as seen in Fig. S4 with the animal 3 (green) example, there can be animal to animal variability, with that particular animal exhibiting rather poor PLR (issue further discussed above and further below). See also Fig S1 as discussed above.

d. Why is the ipsi-contra relationship opposite in Figure 6 when a sinusoidal stimulus is used (Figure 6c and d – contra greater than ipsi). Why do the authors think this is?

In order to titrate the relationship between stimulus frequency, PLR dynamics and PLR-driven retinal activity, we designed two stimuli: a slow (0.55hz) stimulus that would match exactly the pupil kinetics and a faster one (2hz) for which we predicted the PLR to not be able to occur. In the case of the 0.55hz sinusoidal stimulus, the PLR is phase-locked to the stimulus (The PLR starts at the positive amplitude peak and is maximal around the negative amplitude peak of each cycle); in an OFF population, this leads to the PLR-driven response and the canonical OFF response to occur at the very same time for the contra stimulus. Therefore, the responses to the contralateral stimulation presented in figure 6c are a combination of these two different responses (pupil-induced offset and stimulus offset), while the responses to ipsilateral stimulation are simply PLR-driven responses. On the other hand, when stimulating at 2hz, no PLR-driven responses were observed, as the pupil was unable to follow the stimulus, so only canonical OFF responses to contralateral stimulation were observed.

Please note that we also added (Fig. S15B-C) showing the responses to even slower stimuli (0.25hz) where the PLR-driven responses and canonical OFF responses are clearly separated in each cycle of the stimulus, emphasizing the above point.

5. The latency of the delayed-ON response appears to be different in ON and OFF RGCs (Figure 3). Is this consistent? The timing differences of the responses might also be worth reporting more clearly. Moreover, are delayed-ON responses from binocular stimulation faster than monocular stimulation in both ON and OFF cells (Figure 5)?

This is a similar point to question 3 from reviewer 1: In the new insert to panel F and new panel G of figure 3 we further explore the timing of the ipsi responses. We observed that the suppression of responses in ON boutons coincides with the delayed, non-canonical response in OFF boutons. Both responses correlate with pupil constriction and can be explained by the decrease in the number of photons reaching the retina. Some, especially transient-ON cells, also respond with an increase in activity later, during rebound dilation, when the number of photons reaching the retina is increasing.

Regarding the second point referring to Figure 5, further analysis of responses to pupil constriction to ON and OFF cells (Fig. S1E-F) shows that there is no difference in the onset time of the responses.

6. In Figure 3, ON RGCs clearly show an ipsi delayed-ON response to luminance step at the start of the chirp stimulus. However, in Figure S1, the ipsi stimulus does not appear to drive a delayed-ON response despite the text stating so (lines 114-115). Please clarify.

During further examination of the ON population presented in S1 we observed that we failed to separate a small subpopulation of suppressed-ON responses with opposite response polarity,

resulting in averaging-out of the two effects. We corrected this clustering mistake and replaced the relevant panel in supplementary Figure S1A.

However, while the PLR-driven responses during the Full-Field-Flash experiment are visible, they are less pronounced than those seen during chirp stimulation. This can be explained by different luminance conditions in two experiments. Figure S1 represents responses to 50% contrast change to either eye. During ipsilateral chirp, however, the contralateral eye is viewing mean gray stimulus, while the ipsilateral eye is perceiving changes from black to white, causing maximal contrast change. In Fig. S12-S13 we show that increase in contrast leads to the increase in ipsilateral PLR-driven responses.

7. In Figure 5, contra and both eyes are compared. But ipsi induced responses appear to be larger with the flash stimulus (Figure 2). Is the same binocular potentiation seen when comparing ipsi and both eyes?

Yes. When comparing ipsi and both responses we observe binocular facilitation during low contrast stimulation (see new Fig. S13C, left), while during high contrast stimulation (Fig. S13C, right) ipsi responses are larger than either contra or both responses. However, it is worth noting that while the illumination on the contralateral eye is the same during both and contra stimulation (20 LUX for low-contrast, 65 LUX for high-contrast), they differ during the ipsilateral stimulation (9.5 LUX for both high and low contrast), making the comparison hard to interpret.

8. Were ON RGCs analyzed in the Figure 2 and Figure 5 experiments? If so, do they show similar effects? Moreover, are the cells in the Figure 6 and 7 experiments ON or OFF (or combined)? Is the same binocular facilitation seen in ON RGCs?

The ON RGCs responses to low and high contrast stimulation have been added to Fig. S14. We show how PLR modulates the response profile of the ON populations: Fig. S14 represents the positive correlation of the pupil response and the ON boutons responses. Moreover, consistently to what previously showed for the OFF population, the PLR-driven modulation is more prominent during low versus high-contrast stimuli. However, we observe no binocular facilitation in either case (Fig. S14B-C).

Boutons in figures 6 and 7 are OFF boutons. We also analyzed ON boutons (Fig. S15A), that show no binocular facilitation.

9. The Figure S6 caption states that only the subpopulation seen in Figure S6c was used for further analysis throughout the study. However, the subpopulations in Figure S6a also shows potentiation. Are the same results seen with these other subpopulations?

We report binocular facilitation during low-contrast stimulation and quantify it in example population. These results are also seen in the other OFF population. A major difference between the two populations is the strength of PLR-driven response, which is in the first subpopulation more prominent in all conditions. To further address the reviewer's question, we added Fig. S13 which includes all the OFF boutons, and Fig. S12, which includes two OFF sub-

populations (one of which is the one shown in the main figures) and one ON population obtained from the full-field flash data.

10. In Figure 6, one temporal frequency that the pupil could track was tested (0.55 Hz). Would other slow temporal frequencies produce the same results? Were any others tested?

We also tested frequency 0.25 Hz (figure S15B-C), and PLR-driven responses can be observed. As the half cycle of the stimulus is slower than the time necessary for pupil constriction, PLR-driven and canonical responses of OFF boutons are separated, unlike with the 0.55 Hz stimulus. We observe varied responses, with some boutons exhibiting binocularly facilitated PLR-driven peak and others suppression of the canonical part of the response. Interestingly, ON boutons exhibit strong binocular suppression in this condition. The responses to this slower stimulus are consistent with our larger conclusions on the optimal 0.55Hz stimulus compared to the 2Hz stimulus.

11. In Figure 8, which eye were pupil responses in human subjects measured and which eye was stimulated? The important comparison here would be stimulation of one eye and measurement of the pupil response in the other eye (the main effect of the study), which seems important to add here (or discuss, at the very least). Is the optimal temporal frequency for pupil stimulation the same between mice and humans?

For that figure, in both humans and mice, we presented a full-field flash to both eyes, while recording the pupil of one eye. The aim of this experiment was to compare the kinetics between mice and humans. We showed that in mice the amplitude changes between binocular and monocular stimulation, but there is no difference in the kinetics (Fig. S4E-F). Therefore, since understanding the kinetics was our goal, we decided to study it in only during binocular stimulation for Figures 7 and the new psychophysics of Figure 8.

Regarding the second question on optimal temporal frequency, indeed the optimal temporal frequency for mouse and humans is the same—at 0.55Hz. The human was able to only track the 0.55Hz stimulus, not the 2Hz stimulus (Fig. 7C-E), as quantified by power spectrum analysis (Fig. 7D-E), which indeed is similar between mice and humans (Fig. 7E, blue). This was clarified in the results section related to the figure.

11. ON pathways (photoreceptor-mediated and melanopsin) control the pupil response. Would a similar inhibition of the PLR response occur with APB application (blocking ON pathways)? If the delayed-ON response signals a global state change in luminance, is melanopsin involved?

We believe that inhibition of the ON pathway with APB would not bring conclusive results. This is because multiple studies have shown that while injection of APB can reduce the PLR (Sekaran et al. 2007, Young and Kimura 2007), however, it does not abolish the PLR completely. In the work by Sekaran and colleagues (2007) they showed that after application of APB, the pupil still constricts about $\frac{1}{3}$ as much as in control. This is roughly the level of constriction we observe during consensual PLR compared to the direct PLR (Fig. 2G). Still, consensual PLR can reliably drive visual responses. We believe that the use of atropine or

carbachol can act as a more convincing negative control, as it leads to the complete abolishment of PLR and inhibition of PLR driven responses. Indeed we predict that melanopsin is strongly involved in the PLR driven responses, since ipRGCs are the main retinal input to the olivary pretectal nucleus, hence crucial for driving the PLR.

12. Many details concerning the anesthesia/awake status, numbers of mice or recordings per result, and information regarding which boutons were analyzed are missing and should be included throughout the results and in the figure captions. Just to state one example, in the paragraph starting at lines 168 and 195, and in the Figure 4 caption. Please double check this information throughout the paper and report the relevant details so readers can fully interpret the experimental conditions, sample sizes, etc. for each result.

We added the required information throughout the text and the figures. For anaesthesia status, all recordings are anesthetized unless indicated they are in the awake—but we clarified this in the results.

13. Further, it would be helpful to provide more details about the bouton analysis – are these from different RGCs, or the same? Can this be disambiguated (e.g., using trial-by-trial variability / correlations)? It seems important to clarify and potentially correct if this poses statistical repeated measures (i.e., all 8 OFF boutons in a FOV are from the same RGC, versus 8 different RGCs).

For the data presented all over the manuscript, no direct estimation on the number of RGC we imaged was performed. However, under the reviewer's suggestion, we performed a noise correlation analysis on the spontaneous activity recorded from 3 FOVs in 2 different animals, with the same recording parameters as the one used for the data presented in the manuscript. The noise correlation analysis was performed based on the approach proposed by (Liang Liang et al. 2018) and shown in Fig. S7). Similar to their results, we found a median number of boutons per axon ranging from 2.0 to 6.5, the previous study using a median number of 5. We therefore followed the reviewer's suggestion and applied Bonferroni correction to all the p-values reported in the manuscript according to these results, using the most stringent correction of 6.5 boutons per axon. Even after this correction, our reported effects are still overwhelmingly significant.

14. Related to the above, it would be highly beneficial to report how statistically reliable the effects are within subject. For example, is the strength / frequency of “delayed ON responses” similar across mice? Across experiments within a mouse? Such details would enhance the interpretability of findings and illustrate their generality (or not).

New supplemental figure 4 now reports animal to animal variability for 5 animals, showing that difference in the dynamics of the pupil responses can affect also the amplitude and time course of the boutons responses. We hypothesise this form of variability in our dataset to be mainly attributed to the effect of anesthesia, which has been widely shown to affect the speed and the amplitude of light-induced pupil responses (Solessio et al., 2012 IOVS). Even not correcting for this in our data (e.g. even including animals that had suboptimal PLR under anesthesia) the effects were overwhelmingly statistically significant. For instance, our claim

that “between 27% and 55% of RGCs” exhibit PLR-induced binocularity is likely an understatement, since these percentages include animals with poor PLR. This has been mentioned in the text to underscore this notion.

Minor:

15. Figure 1. In panels E and F, the simultaneous pupil traces should be included. Are these examples just one bouton or an average of many boutons?

We added the pupil traces in the figure as suggested by the reviewer. These are single bouton responses, with extra label added to the figure for clarity (Population averages are presented in figure 2-G).

16. Figure 2g. Please plot the ipsi and contra pupil traces on the same timescale – the ipsi response is much smaller. Consider also for Figure 6c and d.

We corrected the amplitude scale of the pupil traces as suggested by the reviewer in both figures.

17. What flash intensity was used in Figures 1 and 2?

Figure 1 E,F and figure 2 G-M are responses to flash ranging from 9.5 LUX to 31 LUX. Figure 2 A-E are responses to a flash ranging from 9.5 LUX to 65 LUX. We added this information in the caption of the figures.

18. Figures 3 and 4. Please include the pupil traces for panels C and D.

Added to supplemental figures 9 and 11 for other recordings, since we do not have paired pupil data for that specific recording in figure 3 and figure S11 (previously Fig. 4). However, the presented response of the pupil to the chirp illustrates how the chirp induces PLR only during the sustained flash stimuli, not the amplitude or frequency ramps. V1 cell body recordings (new Fig. 4F-N), however, do include the pupil.

19. What was the ambient brightness of the experimental room (i.e. were both rod and cone pathways active)? Depending on the RGCs labeled, adaptation state may influence the measured retinal activity.

All the recordings were performed with the lights off but not in complete darkness--not enough to establish full dark adaptation for rod vision, which requires very stringent measures to eliminate all possible stray light from screens, small computer LEDs, etc. Regardless, most of our experiments had a baseline of a grey screen, and the animal was well adapted to that light level, for at least 5 minutes with the screen on before recordings started, which was well in the range of mesopic vision (mixed rod and cone vision). This adaptation largely removed the effect of the room's much dimmer ambient light. That this period of adaptation occurs has been added to the methods.

20. The last paragraph in the discussion suggests this could be related to a global state-change signal to the brain. How do the authors think this increase in activity in OFF

RGCs during a light flash would lead to this effect? Are there other naturalistic stimulus conditions that could induce these delayed-ON responses and binocular facilitation?

Indeed, we briefly speculate on situations where this may occur, with the added line in the discussion "This could be useful when the average luminance on the retina changes, for example, when a shadow falls on a light source (a leaf above moving, blocking sunlight)." Furthermore, one study found that the human pupil engages in frequent periods of contraction and dilation when walking through the woods (Matthis et al., 2022, ref. 17), suggesting this modulation could be rather frequent. Furthermore, we also speculate in a previous paragraph that arousal might cause such state-change signals, overlaid on naturalistic conditions where arousal may elicit a differential response.

21. In Fig. S4, there appears to be a mismatch in the timing of the chirp stimulus schematic at the top and the pupil traces (e.g. the second pupil constriction event begins before the luminance increase). Please check or explain.

This has been corrected in previous Fig. S4, now Fig. S9.

22. Line 162. It appears a citation is missing.

Fixed.

23. Lines 316-318. While the SNR values may be consistent, the response latencies of the effects here and binocular summation for reaction times are drastically different (Blake, Martens, Gianfilippo, IOVS 1980). This should be mentioned, or the line of argumentation dropped.

Indeed that is correct, and thus we decided to change the line of argumentation on binocular summation: we only focused on the predictions for SNR, and highlighted some quantitative analogies between the effect we report and the notion of binocular summation as previously reported in the literature. This led to several changes throughout the document results and discussion.

24. Line 332. What is the definition of the "dominant eye" in this context?

We agree that the expression "dominant eye" is confusing in this context, as the observed binocular responses are not a direct effect of integration of the visual streams coming from the two eyes. Historically, the "dominant eye" is the one that has the largest effect of a binocular response (in our case the contralateral eye), but indeed using this term simply adds unnecessary complexity to the paper. We changed this expression with "recorded eye" throughout the manuscript for clarity.

25. In Figure 7 it would be helpful to label the graphs with what they are showing (e.g. top 20%, etc).

We integrated the information into the figure .

26. L.367 – It seems overly speculative to claim that a conserved delay in PLR kinetics (mainly due to stereotyped neuromuscular kinetics) as evidence for a “conserved mechanism” of error correction of luminance changes (which is not explicitly shown in this study). Please revise or clarify these speculations, or simply remove them as they seem misplaced given the findings of the study.

Speculation on that specific point was removed. However, we added new data to support the more general conservation in PLR-induced activity. To show that the presented phenomena is likely conserved in humans, we performed a psychophysics experiment (also described previously). Subjects reported the perception of a negative luminance modulation which aligns well with the PLR dynamics. We hypothesise this perceptual effect to be informed by a PLR-driven modulation in the neural activity throughout the visual pathway consistent with the one characterized in mice in this study, and thus believe that is more substantial grounds for suggesting a conserved presence of PLR-induced retinal activity between mice and humans.

27. Line 277 and Figure 7. What if all boutons were analyzed?

Results from the SNR analysis of all the boutons for previous figure 7 (current Fig. S17) are added to S17, by replacing S17B , which now shows all boutons. The main point about boutons at the earlier peaks reaching an SNR of 1.4 (square root of 2), and that being removed by atropine, is still present and very statistically significant.

28. Figure 6 title. “constraint” should be “constrain”.

Fixed.

Dear editors of Nature Communications,

Here are my point-by-point response to all reviewer comments:

Reviewer #1 (Remarks to the Author):

I appreciate the authors' additional experiments, revisions to the manuscript, and detailed responses to the reviewers. Although the visual response induced by PLR does not show significant physiological meaning in human psychophysical experiments, this article currently presents a complete and systematic scientific story.

We thank this reviewer (and co-reviewer #2) for their comments in assessing this article is a complete and systematic story, and by not asking for further experiments or clarification, we assume they request no further response before publication. Regarding the human psychophysical experiments not showing physiological meaning, indeed we do not claim any, for instance what this illusion could mean for natural, ethological vision, only that the illusion can be perceived at all, which is emphasized in our discussion.

Reviewer #2 (Remarks to the Author):

Reviewer #3 (Remarks to the Author):

The authors have done a significant amount of work to address my concerns, along with addressing related concerns from other Reviewers. The overall conclusions are strengthened by the additional experiments about background luminance dependence (S3), and simulated dimming with dilated pupil (S8). The line of argumentation focusing on low contrast SNR improvements with binocular facilitation is also stronger (S17). The addition of animal-by-animal variability data are helpful for readers to judge the effects for themselves. Overall, the authors are commended for their improved rigor and stronger support for the conclusions of this very interesting study highlighting novel aspects of binocular PLR interactions.

We thank this reviewer (and co-reviewer #4) for their kind assessment on the rigor and stronger support to our conclusions that we wished to provide with our experimental and textual revisions. We assume the lack of asking for further clarification, modification, or additional experiments means that no further response is required before publication.

Reviewer #4 (Remarks to the Author):

Sincerely,